
# The trajectory of landcover change in peatland complexes with discontinuous permafrost, northwestern Canada

Olivia Carpino[1*], Kristine Haynes[1], Ryan Connon[2], James Craig[3], Élise Devoie[3] and William Quinton[1]

[1]*Cold Regions Research Centre, Wilfrid Laurier University, Waterloo, Ontario, N2L 3C5, CANADA*
[2]*Environment and Natural Resources, Government of the Northwest Territories, Yellowknife, Northwest Territories, X1A 2L9, CANADA*
[3]*Department of Civil and Environmental Engineering, University of Waterloo, Waterloo, Ontario, N2L 3G1, CANADA*
*Corresponding author contact information: ocarpino@wlu.ca (519-884-1970)*

**Abstract**

       The discontinuous permafrost zone is undergoing rapid transformation as a result of unprecedented permafrost thaw brought on by circumpolar climate warming. Rapid climate warming over recent decades has significantly decreased the area underlain by permafrost in peatland complexes. It has catalyzed extensive landscape transitions in the Taiga Plains of northwestern Canada, transforming forest-dominated landscapes to those that are wetland-dominated. The high rate and large spatial extent of this thaw-induced landcover transformation indicates that this region is particularly sensitive to warming temperatures and will continue to respond to climatic changes and landscape disturbances. This study explores the current trajectory of landcover change across a 300,000 km² region of northwestern Canada's discontinuous permafrost zone by presenting a space-for-time substitution that capitalizes on the region's 600 km latitudinal span. To illustrate this trajectory of change we present the distribution of peatland-rich environments that govern permafrost coverage in this region of the discontinuous permafrost zone. We also establish that relatively undisturbed forested plateau-wetland complexes dominate the region's higher latitudes, forest-wetland patchworks are most prevalent at the medial latitudes, and forested peatlands are increasingly present across lower latitudes, indicating not only a climatic gradient but also a landscape in transition as local mean temperatures increase. This study combines extensive geomatics data with ground-based meteorological and hydrological measurements to inform a new conceptual model of landscape evolution that accounts for the observed patterns of permafrost thaw-induced landcover change, and provides a basis for predicting future changes.

**Keywords:** discontinuous permafrost zone; Taiga Plains; peatland; climate change; boreal forest; hydrology; energy dynamics

**Key Points**
1. Geomatics methods are used to generate a distribution of peatland-dominated landscapes in the discontinuous permafrost zone
2. A conceptual model presents landscape evolution in peatland complexes following permafrost thaw
3. A space-for-time approach extrapolates landcover, energy, and water balance field data from the Scotty Creek Research Station to the regional landscape change anticipated across the Taiga Plains.





## 1. Introduction

Arctic and subarctic regions are undergoing unprecedented rates of climate warming and

as a result, these regions are experiencing widespread permafrost thaw (Overland et al. 2019).

Permafrost thaw is one of the most dramatic manifestations of climate warming and has the

potential to drastically change the biophysical features of the land surface. The rate, pattern, and

subsequent stages of thaw-driven landcover change across the circumpolar north are not well

understood. As a result, land and water resources in these regions have uncertain futures. This is

particularly evident across the discontinuous permafrost zone where substantial changes to

landcover (Quinton et al. 2011; Chasmer and Hopkinson 2017) and regional hydrology (Connon

et al. 2014; Korosi et al. 2017; Walvoord et al. 2019) have been documented.

While permafrost (*i.e.* perennially cryotic ground) underlies 16% of the Earth's land

surface (Tarnocai 2009), it is estimated that 80% of the world's boreal forest lies within this

circumpolar permafrost zone (Helbig et al. 2016a). In the southern extensive discontinuous and

sporadic discontinuous permafrost zones in the Taiga Plains ecoregion of northwestern Canada,

permafrost is preferentially located in low-lying, peatland-dominated areas. Such areas are

typically composed of raised, black spruce (*Picea mariana*) covered peat plateaus overlying thin

(<10 m), ice-rich permafrost; and permafrost-free treeless wetlands, including channel fens and

collapse scars, the latter of which result from thermokarst erosion of peat plateaus (Zoltai &

Tarnocai 1975; Robinson 2002; Carpino et al. 2018). The peat plateaus and collapsed wetlands

are arranged into distinct "plateau-wetland complexes" separated by channel fens.

Permafrost temperatures in the Taiga Plains have been warming steadily over the last

several decades (Kokelj et al. 2017; Holloway & Lewkowiz 2019). While permafrost throughout

the southern discontinuous permafrost zones is preferentially located in areas of high peatland



coverage due to the insulating properties of peat (Camill 1999), the observed rapid increases in

air temperature, and consequently ground temperature, have initiated and accelerated permafrost

thaw (Overland et al. 2019; Schuur, 2019). Specifically, the frozen ground along the

southernmost boundary of the sporadic discontinuous permafrost zone is already at or very near

to the 0°C freezing point (Kwong & Gan 1994); indicating a state of disequilibrium with the

current climate (Helbig et al. 2016a). While this condition occurs predominantly along the

southern margin of permafrost, permafrost throughout the discontinuous zone can also be

similarly warm (*i.e.* > -1°C) and vulnerable to thaw and degradation, particularly when

vegetation is disturbed by anthropogenic (Smith et al. 2008; Smith & Riseborough 2010) or

natural (*e.g.* fire) causes (Gibson et al. 2018). Further north, permafrost in the extensive

discontinuous zone is warming, but has not been documented to reach the temperatures found

further south (Smith et al. 2005). While the northern portion of extensive discontinuous

permafrost is not as immediately vulnerable to the thaw and degradation widely documented

across the southern discontinuous zones, the conditions that occur there may provide insight into

the future conditions of northern environments as pan-Arctic warming continues.

The thaw of permafrost below plateaus is driven horizontally by conduction and

advection from the adjacent wetlands, and vertically by conduction from the ground heat flux

(Kurylyk et al. 2016). As permafrost thaws, the overlying plateau ground surface subsides and is

engulfed by the surrounding wetlands (Helbig et al. 2016a). As such, permafrost thaw transforms

forested landcovers into treeless, permafrost-free wetlands (Quinton et al. 2011; Baltzer et al.

2014; Carpino et al. 2018). Zoltai (1993) envisioned this permafrost loss as part of a cyclical

process that also includes regrowth, suggesting that the collapse scar wetland is both the end-

point of permafrost loss and the starting point of permafrost regrowth. This cyclical progression





has now been disrupted by climate warming such that rates of permafrost loss greatly exceed

those of permafrost growth (Robinson & Moore 2000; Jorgenson et al. 2010; Chasmer &

Hopkinson 2017). However, the absence of permafrost regrowth does not preclude the re-

establishment of a black spruce forest (Haynes et al. under review; Chasmer & Hopkinson 2017).

Collapse scar wetlands have been found to revert to such a forest within two or three decades

where the raised permafrost on the wetland margins thaws, allowing the wetland to partially

drain, which is sufficient to allow for root establishment (Haynes et al. 2018; Haynes et al. under

review).

In the Taiga Plains between 55.5° N and 64.6° N, Beilman and Robinson (2003) reported

a 10 to 51% reduction in peat plateaus and thus, the area underlain by permafrost over 50 years.

In recent decades, accelerated thaw rates (Overland et al. 2019; Schuur, 2019; Biskaborn et al.

2019) have fragmented landcovers, profoundly impacting the flux and storage of water (Connon

et al. 2015) and energy (Kurylyk et al. 2016; Devoie et al. 2019). Presently, thermokarst

wetlands occupy more than 60% of the landscape in much of the peatland dominated lowlands of

the Taiga Plains (Olefeldt et al. 2016). Permafrost thaw is generally expected to intensify the

hydrological cycle of high latitude drainage basins (DeAngelis et al. 2015; Box et al. 2019).

Since peat plateaus, collapsed wetlands and channel fens are known to have contrasting

hydrological functions (Hayashi et al. 2004), the permafrost thaw driven change to their relative

cover (Quinton et al. 2011) combined with increased hydrological connectivity of the landscape

(Connon et al. 2015) has been documented to alter the flux and storage of water over the

landscape. Moreover, the reduction in the areal cover of forested plateaus and concomitant

increase in tree-free wetland cover alters ground surface energy partitioning (Kurylyk et al. 2016;

Devoie et al. 2019). The nature of these changes to a landscape's energy balance is governed by

properties of the subsurface, ground surface and the overlying canopy, all of which change as a



plateau transitions to a wetland (Helbig et al. 2016b). For example, insolation at the ground

surface of a mature conifer canopy is roughly one order of magnitude less than for an open

ground surface (Pomeroy et al. 2003) such as a treeless wetland. The low albedo surfaces of the

trunks, branches and stems on plateaus receive significant net shortwave, resulting in relatively

high energy contributions of long-wave emission and sensible heat advection to the plateau

ground surface compared to adjacent wetland surfaces (Helbig et al. 2016b).

This study examines peat plateau-wetland complexes along a latitudinal gradient through

the Taiga Plains in order to improve our understanding of permafrost thaw-driven landcover

change in this region as well as advance our ability to predict changes over the coming decades.

In light of this goal, the specific objectives of this work are to: (1) delineate the current extent of

peatlands, permafrost, and forest distribution along the latitudinal gradient extending through the

zones of discontinuous permafrost; (2) characterise end-members and intervening stages of

landcover transition; (3) for each stage identified, provide an interpretation of the hydrological

and ground surface energy balance regimes based on twenty years of field studies at the Scotty

Creek Research Station; and (4) present a conceptual model of peatland transition during and

following permafrost thaw.

## 2. Methods

### 2.1 Study Region

Much of northwestern Canada's boreal ecoregion is located within the discontinuous

permafrost zone, which ranges latitudinally from extensive-discontinuous (50-90% areal

permafrost coverage) in the north to sporadic-discontinuous (10-50%) in the south. Within this

region, the Taiga Plains ecozone is comprised of a patchwork of mineral and organic terrain.



This study targets the peat plateau-collapse scar wetland complexes found in areas of high

peatland coverage (Wright et al. 2009; Helbig et al. 2016a). While temperature is the

predominant control on permafrost, the presence of near-surface organic materials can allow

permafrost to exist where mean annual air temperatures (MAATs) near, or even exceed 0°C, due

to the thermal offset between the ground surface and permafrost table created by insulating dry

peat soil layers (Vitt et al. 1994; Camill and Clark 1998). Dry (*i.e.* unsaturated) peat is a highly

effective thermal insulator, and for this reason, permafrost presence in peat plateau-wetland

complexes is largely restricted to peat plateaus (Zoltai & Tarnocai 1975; Hayashi et al. 2004;

Quinton et al. 2009). The areal coverage of permafrost in the discontinuous zone has

significantly decreased in recent decades due to increasing MAATs and has resulted in a shift

towards more wetland-dominated landscapes (Thie, 1974; Robinson and Moore, 2000; Wright et

al. 2009; Quinton et al., 2011; Olefeldt et al. 2016).

The discontinuous permafrost zone of the Taiga Plains ecozone covers 312,000 km$^2$ and

is here divided into the areas corresponding with extensive-discontinuous (151,000 km$^2$) and

sporadic-discontinuous (161,000 km$^2$) classifications (Brown et al., 2002; Figure 1). The region

is bounded by the Taiga Cordillera to the west and Taiga Shield to the east and is characterized

by a dry continental climate with short summers and long, cold winters with MAATs ranging

from -5.5°C to -1.5°C (Vincent et al. 2012). Much like the documented panarctic warming trend

(Overland et al., 2019), MAATs have increased across the Taiga Plains where the region has

experienced warming by as much as 2°C over the past 50 years (1970 – 2019) (Vincent et al.

2012). This is largely due to increases in average winter and spring temperatures of

approximately 3°C over the same time period (Vincent et al. 2012). Mean annual precipitation

has largely been consistent in this region over the past 50 years (Mekis & Vincent, 2011).

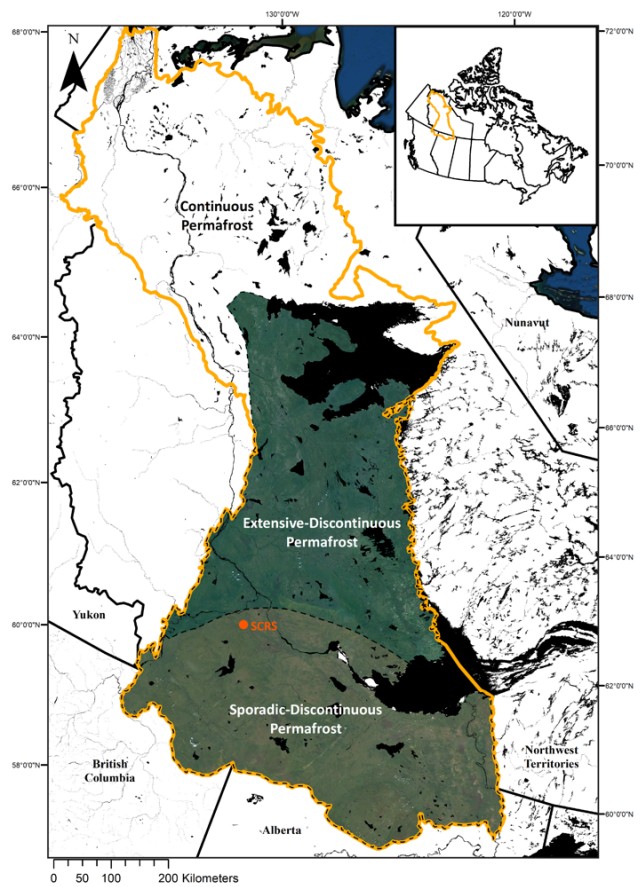

Figure 1: The Taiga Plains ecoregion with the discontinuous permafrost zones (coloured)
defining the study region. The location of Scotty Creek Research Station (SCRS) is also
indicated. Contains information licensed under the Open Government Licence – Canada.

### 2.1.1 Scotty Creek, NWT

The Scotty Creek Research Station (SCRS; 61.3°N, 121.3°W) has been the site of
extensive field-based studies over the past 20 years and provides an opportunity to use long-term
and detailed datasets that are uncommon in northern research (Quinton et al. 2019). The
trajectory of change proposed in this study is based upon observational data collected by the
SCRS and geospatial data across the broader study region. SCRS is located near the confluence
of the Mackenzie and Liard Rivers, and is approximately 50 km south of Fort Simpson in the



Northwest Territories (Figure 1). The MAAT (1970-2015) in Fort Simpson is -2.6°C, with a

mean annual precipitation (1970-2015) of 400 mm, of which mean annual snowfall accounts for

150 mm (Environment and Climate Change Canada 2019). Much like the broader Taiga Plains

region, temperatures in Fort Simpson have been steadily increasing, particularly during the

winter months (Vincent et al. 2015). Scotty Creek drains a 152 km² area dominated by peatlands

ranging in thickness between 2 and 8 m overlying a clay and silt rich glacial till (Quinton et al.,

2019). The peatland portion of the landscape is represented by peat plateau-collapse scar wetland

complexes, where permafrost predominantly occurs below the forested plateau features, while

the surrounding wetlands (specifically channel fens and collapse scar bogs) are devoid of

permafrost and are typically treeless (Hayashi et al. 2004; Quinton et al. 2009).

There is growing evidence that plateau-wetland complexes within peatland-dominated

environments are highly susceptible to shifts in the direction and magnitude of water and energy

fluxes in response to climatic changes (St. Jacques & Sauchyn 2009; Quinton et al. 2019). The

SCRS presents a unique opportunity to study warming-induced landcover changes to plateau-

wetland complexes in the Taiga Plains given the relatively long record of field and modelling

studies at this site, which have coincided with a period of drastic climate warming. Long-term

observations from field research and monitoring by the SCRS facilitates the examination of the

impacts of climate change on peat plateau and collapse scar wetland-dominated landscapes,

which are not only found extensively throughout northwestern Canada but also across the global

subarctic (Olefeldt et al. 2016).

**2.2    Geomatics Methods**

Geomatics methods were used to estimate the current distribution of peatlands, forest, and

permafrost across the study area. The discontinuous permafrost portion of the Taiga Plains


ecozone (depicted in Figure 1) was selected as the boundary for the regional scale geomatics

work completed in this study. Multispectral Landsat 8 imagery (30 m resolution) was acquired

across an area of over 300,000 km$^2$ totalling 70 Landsat scenes. Of these, 59 scenes were used to

construct the base of the mosaic and 11 were used as secondary data to patch and minimize cloud

cover. The 59 primary scenes were acquired in 2017 and 2018 while the 11 secondary scenes

were acquired between 2013 and 2016 as data of suitable quality was unavailable during the

preferred time period. Warm-season image acquisition was selected to prioritize snow-free

scenes and to minimize seasonal variations in soil moisture that can also alter surface albedo

particularly near wetland boundaries (Chasmer et al. 2010). As such, all 70 Landsat tiles were

acquired in June, July, or August, rendering the images seasonally comparable and allowing for a

more streamlined mosaicking process. A colour infrared mosaic (Landsat 8 bands 5, 4, 3

displayed as R, G, B) was created across the study region in ArcGIS (ESRI, Redlands,

California) using a Lambert Conformal Conic projection. The mosaic dataset was colour

balanced and the boundary was amended to the Taiga Plains ecozone including the delineations

dividing the sporadic and extensive discontinuous zones (Brown et al. 2002).

To determine the current distribution of plateau-wetland complexes and related

permafrost, the Landsat mosaic dataset was combined with two complementary products using

the ArcGIS suite of programs. First, a saturated soils dataset (Natural Resources Canada 2017)

was selected to isolate areas that were wetland-dominated and likely representative of the

plateau-wetland complexes targeted in this study. Next, the Northern Circumpolar Soil Carbon

Database (NCSCD) (Bolin Centre for Climate Research 2013) was selected to identify peatlands

within the highlighted wetland-dominated areas. The two datasets were then masked to the Taiga

Plains study region and combined in ArcGIS. The resultant product was mapped to display

peatland terrain across the study region.

An unsupervised landcover classification was subsequently completed across the areas identified by the saturated soils and NCSCD datasets to identify and classify the landcovers within peat plateau-wetland complexes. The first iteration of the unsupervised classification (Iso

Cluster classification approach) targeted 50-75 classes (72 created). The original 72 classes were then aggregated into 12 final classes within the peatland terrain outlined across the Taiga Plains study region. The final 12 aggregated classes include: coniferous (dense and sparse), mixed (dense and sparse), and broad leaf forests stands (dense and sparse), bog, fen, open water, bare ground, cloud, and cloud shadow. The results of this classification were used in combination

with the map of peatland distribution in order to identify the forested landcovers within this broader terrain type. Forested peatlands are particularly indicative of landscape change in this region (Quinton et al. 2010; Baltzer et al. 2014; Chasmer & Hopkinson 2017) and as such, the proportion of coniferous forested area within the total peatland area was quantified across the region's latitudinal span. Proportional coniferous forested area was selected rather than total

forested area to account for the observed spatial differences in peatland distribution across the Taiga Plains. For each degree of latitude, a bin was created for proportional forested area and the median was calculated alongside upper (*i.e.* 75th percentile) and lower (*i.e.* 25th percentile) quartiles. This data was plotted as a function of latitude across the study region. This generated a spatially distributed dataset of forest cover across the peatland-dominated regions of interest that

was subsequently complemented by field data collected by the SCRS to guide the proposed conceptual model.

### 2.3    Field-based Methods

Intensive field studies first began at Scotty Creek in the 1990s with the goal of better

understanding northern peatland landscapes (Quinton et al. 2019). A comprehensive archive of



ground-based energy and water measurements was used in this study to examine the temporal

variation in landscape characteristics at Scotty Creek. These measurements helped to inform a

conceptual model that simultaneously describe both the evolution of peat plateau-collapse scar

wetland complexes subject to a warming climate and also the landscape transition from north-to-

south across the latitudinal and climatic gradient of this study. With each transitional stage of

permafrost thaw-induced landcover change occurring in localized areas throughout the Scotty

Creek basin and its location near the mid-latitude of the discontinuous permafrost zone, the

conceptual model developed using SCRS data also intends to act as a microcosm of the broader

landscape change occurring across the Taiga Plains.

Field research at Scotty Creek has improved the understanding of plateau-wetland

complexes that not only dominate the headwaters of the Scotty Creek watershed but also much

of the Taiga Plains ecoregion (Quinton et al. 2019). Collectively, these studies have contributed

to establishing an understanding of the form and hydrological function of the major landcover

types (*i.e.* permafrost plateau, collapse scar bog, and channel fen). These studies have also

demonstrated how these functions are changing with permafrost thaw (Quinton et al. 2019). In

this study, three components of the hydrological cycle were selected to demonstrate this change:

runoff, evapotranspiration, and storage. Precipitation data were also collected by the SCRS for

use in the water balance portion of this study. Interannual precipitation data was collected with a

Geonor precipitation gauge (Model T200B), which was installed in 2008. The Geonor data

include both rain and snow measurements logged at 30 minute intervals (Table 1). Despite no

recorded changes in precipitation logged by the SCRS, in recent years Connon et al. (2014) and

Haynes et al. (2018) have documented increases in runoff in the Scotty Creek watershed and

adjacent watersheds with longer (~ 40 year) hydrometric records. It is suggested this could be

attributed to increases in wetland connectivity due to permafrost thaw-induced landscape change.





Runoff (mm year$^{-1}$) between 1996 and 2012 was reported in Connon et al. (2014) and extended

to 2017 by Haynes et al. (2018), and is used in the runoff component of the conceptual model

presented here (Table 1).

Table 1: Annual precipitation (2008-2019), runoff (1996-2017), evapotranspiration
(2013-2016), and residual storage values are presented (mm year$^{-1}$) for two distinct
transitional landscape stages at Scotty Creek: a landscape dominated by forest and a
patchwork landscape of near-equal forest and treeless wetland landcovers. Both of these
landscapes represent transitional stages that Scotty Creek has undergone, where more
forest-dominated landscapes are more stable whereas increasing wetland presence can be
seen as an indicator of rapid permafrost thaw.

|  | FOREST > WETLAND | FOREST ≈ WETLAND |
| --- | --- | --- |
| PRECIPITATION | 493 | 493 |
| RUNOFF | 149 | 215 |
| EVAPOTRANSPIRATION | 206 | 255 |
| RESIDUAL STORAGE | 138 | 23 |


Evapotranspiration has also been recently studied at the SCRS by Warren et al. (2018),

who reported evapotranspiration for forests, bogs, and the integrated landscape within the Scotty

Creek watershed between 2013 and 2016. Warren et al. (2018) described the variability in

evapotranspiration for these landcovers and quantified the minimal contribution of

evapotranspiration in forest-dominated landscapes due to the poor transpiration of black spruce

vegetation. Warren et al. (2018) reported daily evapotranspiration values (mm day$^{-1}$), which were

then converted to annual evapotranspiration (mm year$^{-1}$) for this study. These annual values were

used in the evapotranspiration component of the conceptual model in this study (Table 1).

Storage was calculated as the residual of precipitation inputs and evapotranspiration and runoff

outputs for this conceptual model (Table 1). Annual runoff, evapotranspiration, and storage were

analyzed to determine the relative trends in the water balance that are likely to occur across the

proposed trajectory of change (assuming unchanging annual precipitation).

Imagery of the Scotty Creek basin was also utilized to quantify and plot the changing

landcover patterns across the proposed trajectory. The SCRS has acquired aerial photos from

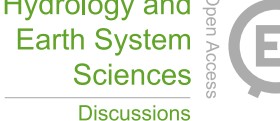

1947, 1970, and 1977 alongside IKONOS satellite imagery from 2000 and Worldview satellite

imagery from 2010 and 2018. The three historical aerial photographs (0.5-1.2 m resolution) and

IKONOS imagery (4 m resolution) were classified and results were presented in Quinton et al.

(2010). Landcover classifications were similarly completed on 2010 Worldview imagery by

Carpino et al. (2018). Additionally, Disher (2020) classified the Scotty Creek basin using the

most recently acquired 2018 Worldview imagery to update the record of permafrost thaw-

induced landcover change. The total area of each landcover (*i.e.* forested plateau, collapse scar

wetland, and afforested wetland) as a proportion of the total area was calculated for both the

historical air photos and more modern satellite imagery at each site. The plotted changes to

landcover proportions between 1947 and 2018 at Scotty Creek were evaluated against the

conceptual model.

Changes to water (*e.g.* Connon et al. 2014; Haynes et al. 2018; Warren et al. 2018) and

landcover (*e.g.* Quinton et al. 2010; Baltzer et al. 2014; Carpino et al. 2018; Disher 2020) due to

climate warming and permafrost thaw have been well documented in previous work by the

SCRS. However, to determine changes to the energy balance across the proposed trajectory, four

meteorological stations at Scotty Creek were selected for use in this study. The first

meteorological station used was installed in a collapse scar wetland in 2004 (hereafter wetland

station) followed by a second station on a densely forested peat plateau in 2007 (hereafter dense

plateau station). Two additional stations located on forested plateaus were also selected to

represent the variety of canopy densities that are increasingly apparent with permafrost thaw-

included landcover change (Chasmer & Hopkinson 2017; Haynes et al. under review). These

stations were installed on a sparsely forested peat plateau in 2015 (hereafter sparse plateau

station) and a forested plateau with an intermediate canopy in 2014 (hereafter intermediate

plateau station).



Four component radiation data were collected at the dense plateau, sparse plateau, and

wetland meteorological stations, while only shortwave radiation was collected at the intermediate

plateau station. Haynes et al. (2019) thoroughly summarize the instrumentation within the Scotty

Creek basin. All radiation data were collected at 30 minute intervals and compiled into daily

averages. The daily average radiation data was used to calculate annual averages across each

station's record length. These annual averages were plotted alongside the proposed trajectory

according to the transitional stage that most appropriately matched the landcover represented by

each meteorological station. While using four meteorological stations (three of which represent

plateau landcovers), accounts for some of the variability present across the landscape, there is

also strong spatial and temporal variability in subcanopy shortwave and longwave radiation

(Webster et al. 2016). To address this more localized subcanopy variability, the daily average

four component radiation data from each station were compared on a monthly time step. The

monthly averages were calculated and compared across the landcovers represented by each of

the four meteorological stations using a one-way analysis of variance (ANOVA) with Tukey

post-hoc test ($\alpha = 0.05$).

Data compiled from the SCRS were used to inform the conceptual model of landscape

transition proposed in this study. The conceptual model presents both the landcover changes

anticipated as an environment experiences permafrost thaw as well as the associated changes to

water and energy. Illustrations were created to represent each stage in the proposed trajectory of

landscape transition. Complementary imagery was collected using a Remotely Piloted Aircraft

System (RPAS) at Scotty Creek to represent how each of these illustrated trajectory stages

manifests on the landscape in a peat plateau and collapse scar wetland-dominated environment.

The RPAS imagery (0.5 m resolution) was collected in the summer of 2018 using an eBee Plus

equipped with a senseFly SODA 3D mapping camera and all image processing was completed in





Pix4DMapper. The changes in energy and water balances identified in literature and data

analysis for this study were calculated according to the landcover and transitional stage

represented by these data. The changes to landcover at Scotty Creek were also presented as a

summary plot within the conceptual model to illustrate the observed and recorded changes over

time. As each of the stages of landscape transition proposed in the conceptual model are also

observed at Scotty Creek, ongoing shifts in landcover and the associated hydrometeorological

changes can be extrapolated to other peat plateau-collapse scar wetland sites similar to Scotty

Creek.

## 3.  Results and Discussion

### 3.1    Peatland and Permafrost Distribution

The geomatics methods applied in this study indicate that the peatland-dominated terrain

often supporting permafrost is distributed across the Taiga Plains (Figure 2). Expectedly, large

peatland clusters are located in lowland areas with high histel soil percentages. In the extensive-

discontinuous permafrost zone, peatlands are clustered to the west, nearest to the Mackenzie

River, and are largely absent from the eastern study region in the area bounded by Great Bear

Lake to the north and the Taiga Shield to the east. In the sporadic-discontinuous zone, the

distribution of these peatland clusters is more longitudinally dispersed. We estimate that

approximately 35% of the discontinuous permafrost zone within the Taiga Plains ecozone is

composed of landscapes with high peatland coverage.

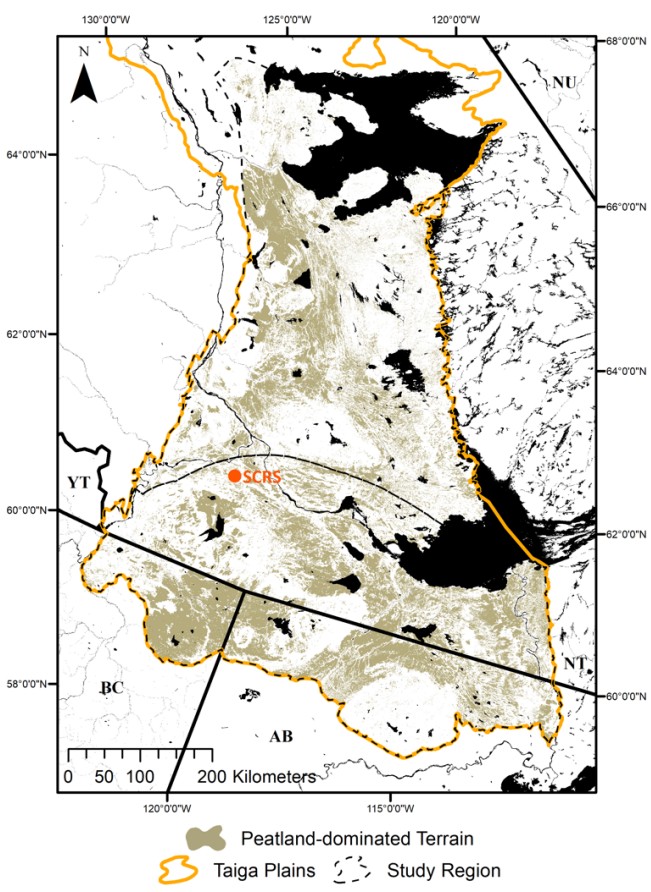


Figure 2: Predicted distribution of peatland-dominated terrain in the discontinuous permafrost zone of the Taiga Plains. Peatland-dominated terrain was determined using a saturated soils dataset (Natural Resources Canada 2017), the NCSCD (Bolin Centre for Climate Research 2013), and Landsat 8 Data from the United States Geological Survey. Contains information licensed under the Open Government Licence – Canada.

As changes to forested landcovers have been used as an indicator of broader landscape

change in this region (Baltzer et al. 2014; Chasmer & Hopkinson 2017), forested peatlands,

including forested peat plateaus in plateau-wetland complexes and forested permafrost-free

wetlands were plotted as a function of latitude (Figure 3). A latitudinal trend in landcover

percentage is apparent within the identified areas of high peatland coverage. Along the boundary





between the extensive-discontinuous and sporadic-discontinuous permafrost zones in the centre

of the study region, wetland features, including collapse scar bogs, are most prevalent. Median

proportional forest cover reaches its minimum at ~ 33% within the 61°N bin, the latitude at

which SCRS is also located. The proportion of forested peatlands remains relatively low across

the transitional zone between sporadic and extensive discontinuous permafrost as median forest

cover does not exceed 34% between 61and 62°N. As collapse scar wetlands appear to be

widespread in this area, permafrost thaw and increased inundation and waterlogging of

previously dry peat plateaus may be most drastic at the mid-latitudes of the study region (Islam

& Macdonald 2004; Iwata et al. 2012). However, in both the extensive discontinuous permafrost

zone to the north and the sporadic discontinuous permafrost zone to the south, relative increases

in proportional forested peatland area are observed.

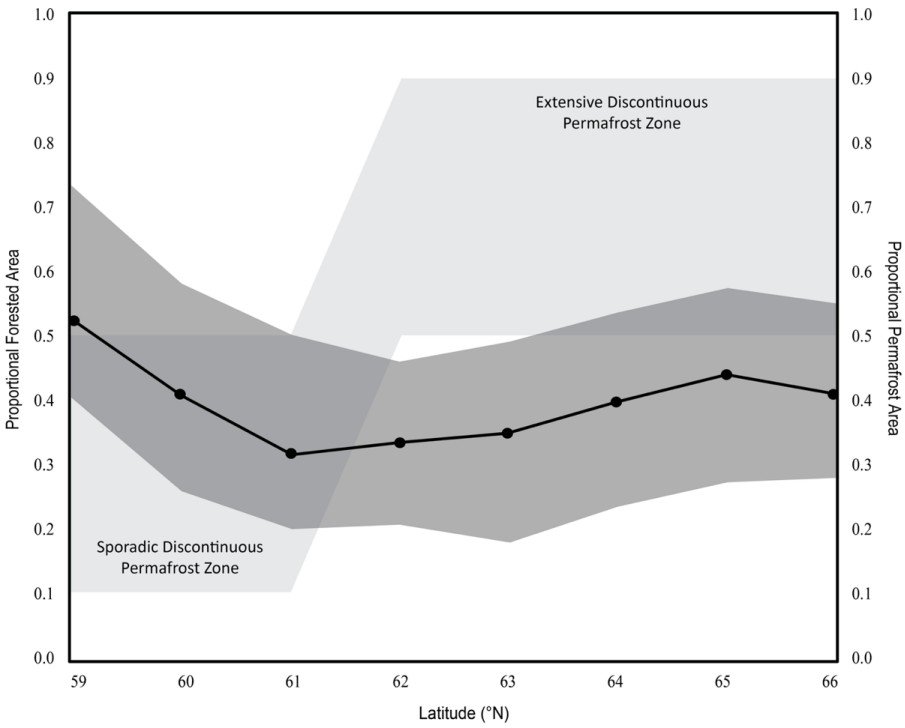


Figure 3: Median forested peatland area proportional to total peatland area plotted as a function of latitude. The extent of the band represents the range of proportional forested peatland area between the 25th percentile and 75th percentile. Permafrost extent as a proportion of total area is also plotted across the same latitudinal range in light gray.


In the extensive-discontinuous permafrost zone (63 − 66°N), the median proportion of

forested peatlands ranges from ∼ 35 − 45% , indicating that forested peatlands, including the peat

plateaus that support permafrost in plateau-wetland complexes, are more intact than across the

transitional boundary zone. This has been explained through lower MAATs and the insulating

properties of the dry peat that dominates the near-surface of these plateau-wetland complexes

(Zoltai & Tarnocai 1975; Hayashi et al. 2004). The sporadic-discontinuous permafrost zone

consists of ∼ 50% (59 − 60°N) forest cover in peatlands, including the greatest median

proportional forested area at ∼ 52%. Here, afforestation of permafrost-free peatlands appears to

be responsible for some of the forested area along the southern boundary of the study region,





particularly in the areas of northern British Columbia and northern Alberta; a pattern also

observed in Carpino et al. (2018), but first reported by Zoltai (1993). This suggests that a large

portion of permafrost has already been lost from these environments as peatland dewatering

lowers the water table (Ketteridge et al. 2013; Haynes et al. 2018), allowing for forest cover to

return to newly unsaturated areas (Zoltai 1993; Camill 1999).

**3.2  Plateau-Wetland Complex Landscape Trajectory**

   The evolution of peat plateau-collapse scar wetland complexes over time can be

represented by seven phases: (I) Forested permafrost plateaus; (II) Forested permafrost plateaus

with small, isolated collapse scar wetlands; (III) Forested permafrost plateaus with larger,

interconnected wetlands; (IV) Wetland complexes with small plateau islands; (V) Wetland

complexes with small-scale hummock development and tree establishment; (VI) Hummock

growth with forest establishment; and (VII) Afforested wetlands (Figure 4). A sequence of

hydrological processes and energetic mechanisms occurs to initiate landscape change and drive

the landscape along the proposed spectrum of transition. As permafrost thaw commences, a

landscape dominated by forested plateaus and underlain by permafrost (Figure 4I) transitions to

one with small, suprapermafrost taliks (Connon et al. 2018) and isolated collapse scar bogs begin

to emerge (Figure 4II; Quinton et al. 2011). As permafrost thaw and talik development

continues, the previously isolated collapse scars expand (Figure 4III; Devoie et al. 2019) and

become interconnected with surrounding wetlands (Connon et al. 2015). As permafrost thaw

continues, wetlands proliferate and become increasingly connected, creating a landscape

dominated by widespread wetland complexes with only isolated plateau islands (Figure 4IV;

Baltzer et al. 2014; Chasmer & Hopkinson 2017).





The predominance of wetland features on the landscape is also coupled with increased and expedited drainage (Connon et al. 2014; Haynes et al. 2018). Over time, small-scale hummock microtopography emerges towards the centre of these draining wetlands (Figure 4V; Haynes et al. under review) and black spruce trees begin to re-establish on these features (Figure 4VI; Iversen et al. 2018; Dymond et al. 2019). Continued hummock growth allows for afforestation to continue (Eppinga et al. 2007; Iversen et al. 2018) until the landscape returns to a forest-dominated landscape that appears similar to the original stage in as little as 40 years (Figure 4VII; Carpino et al. 2018). However, the widespread permafrost aggradation predicted by Zoltai (1993) and Camill (1999) is unlikely to occur under the region's current and warming climatic conditions. Instead, small patches of isolated frozen ground may be able to re-establish under the new black spruce canopy on an interannual basis, but is unlikely to exist at a landscape scale or to persist on a long temporal scale.



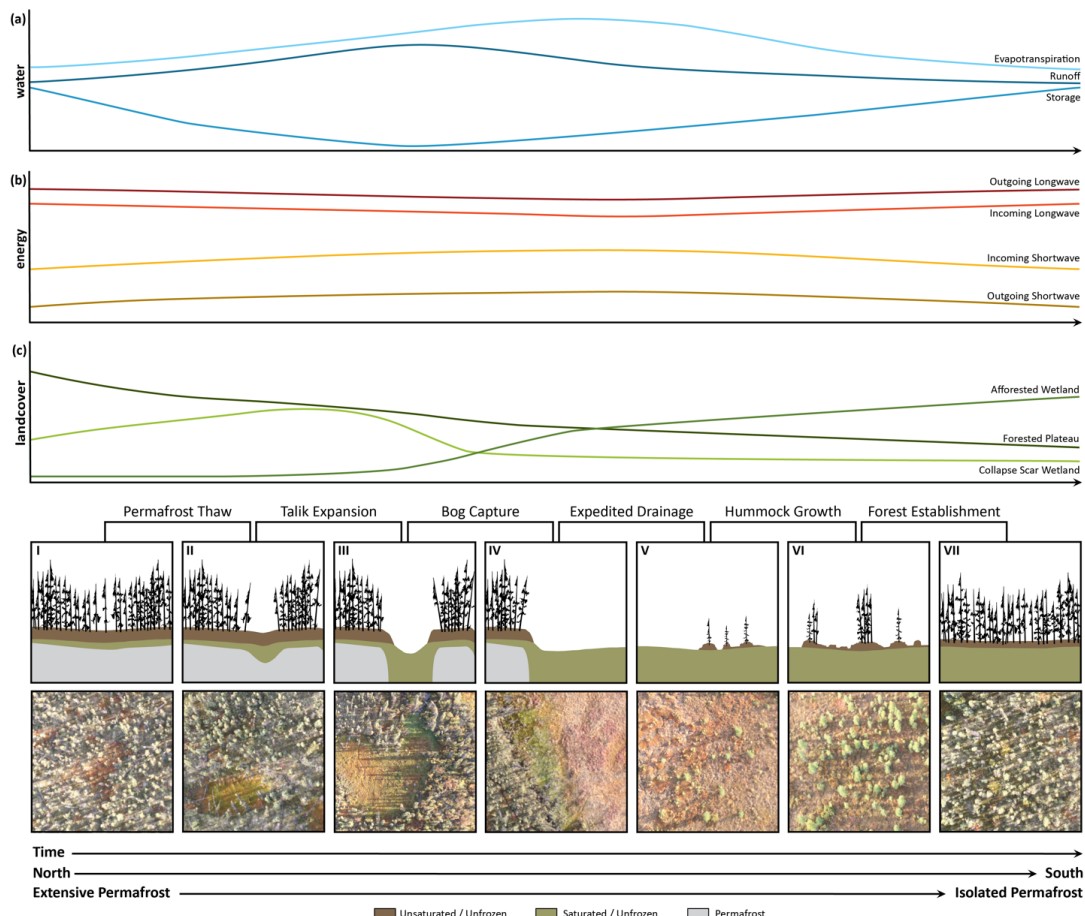

Figure 4: (Bottom) Proposed conceptual model of landscape trajectory including a space-for-time substitution for changes to both permafrost and landcover. Conceptual diagrams are presented to illustrate landscape change with the support of RPAS imagery collected at the SCRS. The conceptual model is presented alongside the processes that initiate the trajectory's progression. (a) Relative changes to local water balances of measured SCRS runoff, evapotranspiration and residual storage with unchanging precipitation are summarized and presented over the trajectory of landscape change based on the proportion of forested vs. wetland area. (b) Relative changes to local energy balances are presented using data collected from meteorological stations installed at Scotty Creek. (c) Changes to relative landcover proportions are presented using historical aerial photographs and recent acquisitions of satellite imagery over the Scotty Creek basin.



This transition not only represents the dominant trajectory of change observed over recent decades in the Scotty Creek basin (Quinton 2019), but also corresponds with the north-south

climatic transition of peatland-dominated plateau-wetland complexes spanning northwestern Canada's zone of discontinuous permafrost. This spans the extensive permafrost found beneath treed peat plateaus in the north, to patchwork or even wetland-dominated landscapes at more moderate latitudes, and finally to more widespread permafrost-free environments at the southern extent of the Taiga Plains. Each of these stages, and the transitional steps between them, can also

be observed at smaller scales at local sites including Scotty Creek, which is located on the boundary between the sporadic-discontinuous and extensive-discontinuous zones. The Scotty Creek Research station is in phase III-IV, with localized conditions that correspond to the range of other phases. This exemplifies the concurrent nature of the phases along the spectrum of landscape transition.

**3.2.1  Landcover Trajectory**

Changes to the relative proportions of the three main landcovers represented across the trajectory (Figure 4c) (forest plateau, collapse scar wetland, and afforested wetland) have been observed at the landscape-scale with remotely-sensed imagery. At Scotty Creek, the early portions of the trajectory (*i.e.* stages I, II) represent the changes observed between 1947 and 2000

as the landscape transitioned from predominantly forested and underlain by permafrost to one defined by a forest-wetland patchwork and degrading permafrost bodies (Quinton et al. 2010). This ~ 50 year period shows a change in proportional landcover as forested areas move from covering approximately 70% of Scotty Creek to approximately 50% (Quinton et al. 2010; Baltzer et al. 2014). This forest loss directly corresponds with wetland expansion at the expense of

permafrost as treeless wetland features (both collapse scar bogs and channel fens) shift from approximately 30% of the landcover to approximately 50% over the same time period (Quinton





et al. 2010). As of 2018, the headwaters of the Scotty Creek basin were comprised of

approximately 40% forested permafrost plateau, 45% treeless wetland (13% collapse scar

wetland and 32% channel fen), and 13% afforested wetland (Disher 2020). These results indicate

that the proportional area of peat plateau, and thus permafrost terrain, has continued to decline

since previous analyses (Quinton et al. 2010; Baltzer et al. 2014; Connon et al. 2014; Carpino et

al. 2018). While permafrost thaw-induced forest loss continues rapidly at Scotty Creek, forest re-

establishment is also occurring in the form of afforested wetlands (Disher 2020). As such, these

results are also indicative of the fact that the transition from one stage of the trajectory to the next

is not instantaneous and can occur to varying degrees within a local site such as Scotty Creek.

Between 1970 and 2010 the Scotty Creek watershed had lost approximately 12% of its

forested permafrost plateau area (Carpino et al. 2018). Landscape wide, the Scotty Creek basin

is, as of 2018, most closely represented by the transition from stage III to IV, where forest and

permafrost loss continue to dominate, but forest re-establishment is becoming increasingly

apparent as the wetlands begin to drain more readily. The transition of forest to wetland is

expected to continue until the later stages of the trajectory, when increases in afforestation will

be observed, yielding decreases in the proportional area of both forested plateaus and collapse

scar wetlands (Ketteridge et al. 2013; Chasmer & Hopkinson 2017; Warren et al. 2018). While

Scotty Creek and many similar sites continue to undergo active transition from plateau to

wetland, afforested wetlands are appearing across the site and may have previously been

misclassified as forested plateau area (Disher 2020). The emergence of afforested wetland

landcovers has previously been documented in sites along the southern margin of sporadic

discontinuous permafrost, particularly in northeastern British Columbia (Carpino et al. 2018).

These sites experienced an even more accelerated transition from forest to wetland and lost close

to 17% of forested plateau area over the same 1970-2010 period (Carpino et al. 2018). However,

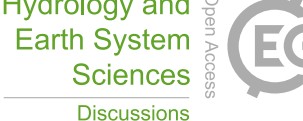

these sites also demonstrated an almost 10% gain in afforested wetland over the same 40-year time period (Carpino et al. 2018).

In addition to establishing the predominant direction of change across the region, the proposed landscape trajectory also corresponds to a succession of plant communities related to

each stage's degree of wetness. Early stages of landscape transition are characterized by the presence of relatively undisturbed black spruce tree cover (Figure 4I). As permafrost thaw progresses, changes to the *Sphagnum*-dominated communities within collapse scar features are an indicator of vegetation succession, demonstrating a wetness-based zonation (Zoltai 1993). Specifically, aquatic *Sphagnum* species, notably *S. riparium*, occupy the inundated margins

between actively thawing permafrost plateaus and developing collapse scar bogs (Garon-Labrecque et al. 2015; Pelletier et al. 2017). Young and expanding collapse scar bogs are most easily identified by the distinct bright green colour of *S. riparium* as seen in high-resolution RPAS imagery (Figure 4II, III, IV; Gibson et al. 2018; Haynes et al. under review) but their margins may also be identified by bare peat banks or moats of water directly along the

permafrost plateau edge (Zoltai 1993). As collapse scars expand, lawn species, such as *S. angustifolium,* and hummock species, such as *S. fuscum*, emerge, particularly towards the bog centre (Zoltai 1993; Camill 1999; Pelletier et al. 2017). Hummock species are especially dominant towards the centre of collapse scars, establishing themselves on top of continuous and sufficiently compacted peat above the water table (Camill 1999; Loisel & Yu 2013). Much like

*S. riparium*, *S. fuscum* is easily identifiable in high-resolution imagery by colour. The distinct russet colour of *S. fuscum,* (Figure 4V) alongside the textured hummock microtopography apparent in high-resolution digital terrain models, provides evidence to aid in classifications (Haynes et al. under review). The abundance of relatively dense *S. fuscum* hummocks allows for the re-establishment of black spruce (Liefers & Rothwell 1987), first on isolated hummocks





(Figure 4VI) but eventually leading to widespread afforestation (Figure 4VII; Camill 2000;

Ketteridge et al. 2013). Ultimately, these successional changes to *Sphagnum* communities

provide clear support to the proposed trajectory, progressing from aquatic-to-lawn-to-hummock

dominated collapse scars before these features are open to forest re-establishment (Iversen et al.

2018; Dymond et al. 2019).

**3.2.2  Energy Trajectory**

The Scotty Creek watershed serves as a microcosm of the transition observed across the

Taiga Plains with respect to their water and energy budgets. While energy budgets measured at

Scotty Creek are expectedly similar across the landscape, some general trends are present (Figure

4b). Annual shortwave radiation, both incoming and outgoing, peaks over the middle stages of

the trajectory (IV, V); where treeless collapse scar bogs are the dominant landcover. Annual

shortwave radiation, both incoming and outgoing is comparatively lower in forested stages of the

trajectory. Specifically, lowest annual shortwave contributions (both incoming and outgoing) are

observed at the dense plateau station (I). Both incoming and outgoing annual longwave radiation

are greatest in the more densely forested landscapes present at the beginning (I, II) and end

stages of the trajectory (VI, VII) and lowest in the wetland landscapes that dominate the middle

stages (IV, V).

Given the spatial and temporal variability of four component radiation data, averages at

each of the four stations representing the different tree canopy densities observed along the

spectrum of landscape change were also compared (Figure 5). Differences between stations were

assessed for statistical significance by one-way ANOVA using monthly data over the length of

record available at each station. There were statistically significant differences between stations

for incoming shortwave (Figure 5a), outgoing shortwave (Figure 5b), and incoming longwave

(Figure 5c), while outgoing longwave showed no statistical differences between stations (Figure





5d). To determine significant differences between each of the landscape types represented by the

four meteorological stations, a Tukey post-hoc test was used for each of the four radiation

components.



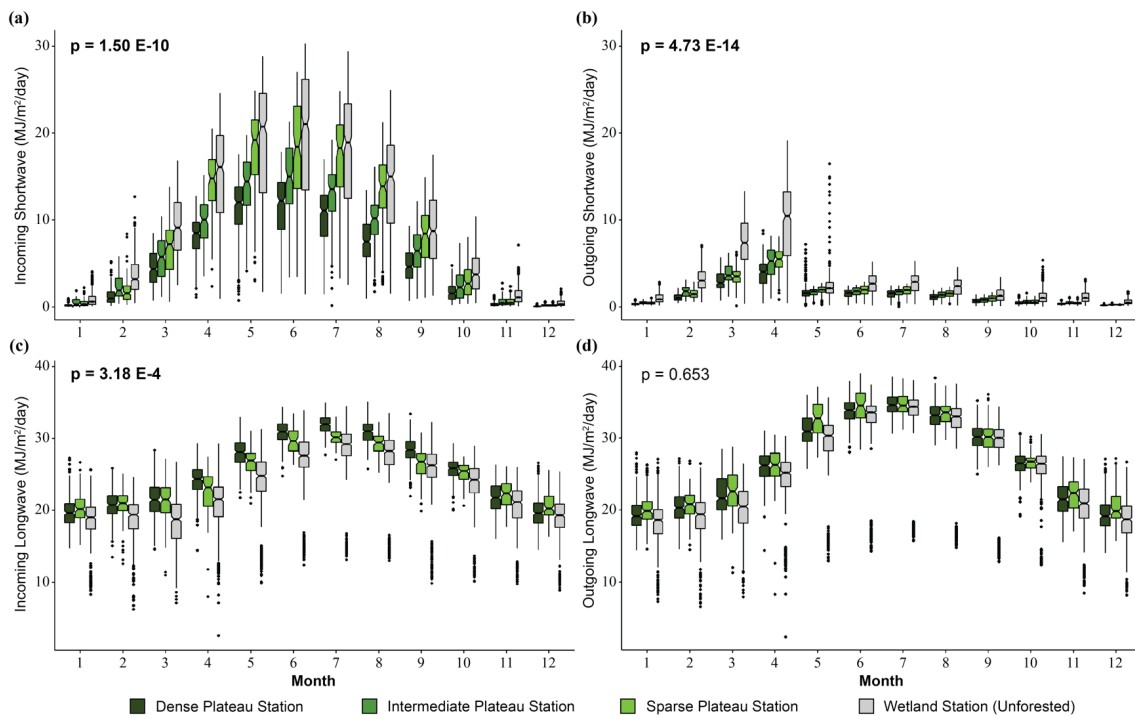

Figure 5: Four component radiation plots (MJ/m²/day) displaying (a) incoming shortwave, (b) outgoing shortwave, (c) incoming longwave, and (d) outgoing longwave across four meteorological stations. Each station represents a distinct landcover: dense plateau (2007-2019), intermediate plateau (2014-2019), sparse plateau (2015-2019), and tree-free wetland (2004-2019). The four meteorological stations are presented as notched box and whisker plots across 12 months where the boxes represent the 25th and 75th percentile, while the whiskers represent the range of the data. The notches on each box indicate the confidence interval (α = 0.05) around the mean while the statistical differences between meteorological stations have been presented in the upper left of each plot as determined by one-way ANOVA. Significant $p$ values have been highlighted with bold text.




Shortwave radiation, both incoming and outgoing, showed significant differences

between landcovers. However, a Tukey post-hoc test revealed that there was variability between

the two shortwave components for which groups specifically showed these significant

differences. As the four meteorological stations fall along a gradient of forest density from a

completely unforested wetland to a densely forested plateau, no significant differences in

incoming shortwave radiation were ever found between stations only one rank apart on that

gradient. As such, measurements indicate average monthly incoming shortwave radiation is

significantly greater at the wetland compared to both the intermediate plateau ($p = 2.82 \times 10^{-4}$)

and the dense plateau ($p = 1.055 \times 10^{-10}$), while no significant difference exists between wetland

and the sparse plateau. On the other end of the gradient, the dense plateau receives significantly

less incoming shortwave radiation than both the wetland ($p = 1.055 \times 10^{-10}$) and the sparse

plateau ($p = 0.036$), but this station is not significantly different from the intermediate plateau.

Outgoing shortwave radiation follows a slightly different pattern as no significant differences

exist between any of the forested plateau stations but all are significantly different from the

wetland. Specifically, outgoing shortwave radiation recorded at the wetland station is

significantly greater than the sparse plateau ($p = 4.5 \times 10^{-5}$), intermediate plateau ($p = 2.0 \times 10^{-5}$),

and dense plateau stations ($p = 4.814 \times 10^{-13}$). The differences between the wetland station and

the forested plateau stations also become progressively more significant with increasing tree

density.  There was a statistically significant difference between the three stations (intermediate

plateau omitted due to lack of measurements) for incoming longwave radiation, while no

statistically significant differences were observed in the outgoing longwave radiation component.

The only significant difference in incoming longwave was observed between the wetland and





dense plateau ($p$ = 2.26 x 10$^{-4}$). No significant differences exist between the sparse plateau and the wetland or dense plateau.

The radiation data plotted follows the patterns that would be expected at a subarctic site such as Scotty Creek, where drastic decreases in available energy are observed in winter months (Figure 5). At sites like Scotty Creek, where a forest-wetland patchwork is present, the observed differences in radiation between the landcovers represented by each meteorological station can also be amplified by temporal variability. Differing surface properties, particularly albedo, at each landcover type can alter the impact of shortwave radiation on the landscape. Wetland

albedo has been observed to be consistently higher than forest albedo at Scotty Creek as black spruce forests cover the more reflective ground cover (*e.g.* mosses, lichen, deciduous shrubs, etc.) (Helbig et al. 2016b). However, the differences in albedo are further exaggerated when snowcover is present as black spruce forests also mask the highly reflective snowcover (Helbig et al. 2016b). As incoming shortwave radiation increases towards the end of winter in the period

leading up to snowmelt, the impact of albedo on outgoing shortwave radiation is particularly prominent. This is evident as outgoing shortwave radiation at the wetland station increases ahead of the forested stations during late winter and then peaks in the spring due to the impact of albedo at the tree-free wetland site (Figure 5b). At sites such as Scotty Creek, where permafrost thaw-induced landcover change is observed as a shift from forest to wetland, an increase in

landscape albedo has been observed (Helbig et al. 2016b). This could lead to a regional cooling effect across the southern Taiga Plains, particularly during winter months when differences in albedo are greatest between forest and wetland landcovers (Helbig et al. 2016b). A warming effect was observed during summer months in areas experiencing active forest loss as a thinning





forest increased the ground heat flux, and is consistent with the ongoing permafrost thaw

observed on forested plateaus at Scotty Creek (Helbig et al. 2016b).

In permafrost terrains, the flux of energy is closely coupled with both changes to

permafrost thaw rates (and therefore changes to landcover) as well as changes to runoff (Quinton

et al 1999). Canopy thinning due to permafrost thaw or other disturbances such as fire or seismic

exploration increases the radiation at the ground surface. This spatially heterogenous increase in

incoming radiant energy may be responsible for areas of accelerated permafrost thaw, resulting

in topographic variation across the frost table. This heterogeneity is suggested as the mechanism

driving the transition from stage I to II in the trajectory (Quinton et al. 2019). The topographic

variability in frost table results in preferential water storage in local depressions, increasing the

thermal conductivity and resulting in increased thaw (Quinton et al. 2019). This positive

feedback leads to the formation of depressions both at the ground surface and at the frost table,

altering local runoff pathways and increasing depressional storage (Wright et al. 2009; Quinton

et al. 2009). This positive feedback mechanism is furthered as increased surface wetness

accelerates canopy loss due to waterlogging and therefore increases both the thermal and

radiative energy received at the ground's surface. This feedback is present in the initial stages of

the trajectory, and is often associated with talik formation and expansion into collapse scars due

to localized permafrost loss (Chasmer & Hopkinson 2017; Connon et al. 2018).

### 3.2.3  Water Trajectory

At peatland-dominated sites such as Scotty Creek, changes to energy can act as a driver

of permafrost thaw-induced landcover change and are also closely coupled to changes in water

and runoff regimes. Therefore, as the trajectory progresses through the proposed stages, the

relative influence of the processes governing the hydrology of the landscape changes



dynamically (Figure 4a). The early stages (I, II) of the proposed successional trajectory are dominated by hydrologically isolated wetlands, where storage is maximized and outflow from the basin is limited to primary runoff from plateau features into nearby fens (Connon et al. 2014;

Quinton et al. 2003). Water arriving directly into one of the isolated, but rapidly expanding collapse scar wetlands will be stored as there is no direct flow path to the channel fen (Quinton et al. 2019). The contributions of evapotranspiration during the forest-dominated stages are similarly minor due to the limited transpirative ability of black spruce on organic soils (Warren et al. 2018). Even when forest is the predominant landcover, understory vegetation is the

principal contributor to evapotranspiration (Chasmer et al. 2011).

As the environment becomes more wetland dominated, runoff and storage diverge. By stage III, where wetlands are interconnected and rapidly expanding, storage reaches its minimum while runoff is maximized (Figure 4a). At this point in the trajectory, elevated runoff is due to the introduction of new runoff pathways (Connon et al. 2014), and from the drainage of water

previously stored in isolated wetlands (Haynes et al. 2018). Specifically, runoff is amplified by enhancing the runoff contributing area through connected and cascading wetlands, which are also responsible for the reduction in storage (Connon et al. 2015; Haynes et al. 2018). A decrease in runoff then follows as plateau loss continues to accelerate, eliminating transient contributions from interior collapse scar wetlands, increasing the basin storage and reducing the impact of

cascading wetlands (Quinton et al. 2019). At this point in the trajectory, the impact of storage and runoff on the overall water balance diminishes as evapotranspiration increases and eventually reaches its maximum. In the trajectory stages where wetland is the predominant landcover (IV, V), the evapotranspiration term peaks due to evaporation from standing water and exposed moss groundcover (Warren et al. 2018).





Finally, the advanced stages of the trajectory (VI, VII) correspond with continued decreases in runoff, as well as decreases in evapotranspiration. Peatland dewatering acts as the initial mechanism for the development of hummock microtopography and tree re-establishment (Ketteridge et al. 2013; Chasmer & Hopkinson 2017). The afforested wetland landcover proposed as the final stage in the trajectory has only recently been defined and studied at Scotty

Creek (Haynes et al. under review; Disher 2020) and as such, evapotranspiration, storage and runoff measurements that are specific to afforested wetlands do not presently exist at the site. Soil moisture across the afforested wetland landcover has been observed to be intermediate between that of plateaus and collapse scar bogs (Haynes et al. under review). It is likely that many of the thermal and hydrological characteristics of afforested wetlands would fall between

collapse scar bogs and forested plateau features as they are treed but permafrost-free. This would likely result in similar evapotranspiration relationships to the forested permafrost plateaus that dominate the early stages despite being permafrost-free. Storage and runoff may however increase from the original forest-dominated landscape to a mid-level between plateau and wetland values due to the lack of underlying permafrost and lower elevation.

Afforested wetland features have not been studied to the same degree as peat plateaus or collapse scar wetlands and in some cases have been misclassified as either feature in the past (Haynes et al. under review). Data that are specific to these features are lacking in comparison to better-documented landforms even at sites such as Scotty Creek where intensive field studies are ongoing. Further examination is warranted into the hydrology and energy dynamics at these

sites, particularly if afforestation represents an end-stage in the trajectory of peat plateau-collapse scar wetland complexes. The impacts of increasingly widespread afforested wetlands on basin runoff should be investigated as well as further work to confirm whether evapotranspiration for





this landcover is intermediary much like soil moisture appears to be (Haynes et al. under review).

Furthermore, the persistence and permanence of afforested wetland landcovers has not been

documented. Exploring canopy or stand properties such as age, health, and productivity across

these features may provide insight and assist in further defining the role of afforested wetlands in

the trajectory of plateau-wetland complexes.

## 4.  Conclusions

The discontinuous permafrost zone of the Taiga Plains exemplifies a landscape in

transition. Coupling a broad-scale mapping initiative with the detail of site-specific data

collected at the SCRS emphasizes the close dependence of landcover on climate by way of local

energy and water budgets. Small changes to climate or tree cover can initiate permafrost thaw

and trigger a series of positive feedbacks related to energy, water and vegetation communities,

leading directly to changes across the landscape. The proposed conceptual model of landscape

evolution summarized in Figure 4 describes the transitions occurring across the Taiga Plains in

peat plateau-collapse scar wetland complexes like Scotty Creek. The evolution model is strongly

supported via both direct observations and synthesis of the literature and describes the shifts that

occur in energy and water budgets as the landscape transitions from forest underlain by

permafrost to permafrost-free afforested wetlands. We identify the likely region of applicability

of this conceptual model across a large region of the Canadian north. We also establish the

regional pattern of change across these environments and given the latitudinal gradient is a

suitable space-for-time proxy, project their future trajectory by combining long-term field

observations with analyses of contemporary and historical imagery.  It is proposed that, while

permafrost thaw-induced landcover changes have previously been dominated by a transition

from forest to wetland, this transition is not permanent and forested landcovers are likely to





return over time, although not likely underlain by permafrost. This research improves our understanding of how changes to permafrost distribution and ongoing permafrost thaw may impact peatland-dominated environments and are of relevance to other peatland-rich permafrost environments across the circumpolar north.

**5. Acknowledgements**

We gratefully acknowledge the support of the Dehcho First Nations, in particular, the Liidlii Kue First Nation and Jean Marie River First Nation. We also thank these communities for their long-standing support of the Scotty Creek Research Station. This work was funded by ArcticNet through their support of the Dehcho Collaborative on Permafrost (DCoP), and by the

Natural Sciences and Engineering Research Council of Canada (NSERC). We also acknowledge the Canada Foundation for Innovation (CFI) for providing funding for infrastructure critical to this study.

**6. Data Availability**

The data used in this paper are in the process of being catalogued for open access in the Wilfrid

Laurier University (WLU) Data Repository, which is fully in compliance with all FAIR guidelines. While these data are being catalogued, the datasets used in this study are available upon request by contacting the corresponding author.

**7. Author Contributions**

All authors contributed to the development of the research question and the methodological

approach used in this study. OC performed the analyses and wrote the manuscript with input and editorial contributions from KH, RC, JC, ÉD and WQ.





## 8. Competing Interests

The authors declare that they have no conflict of interest.

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
