# Peer review of "Long-term climate-influenced land cover change in discontinuous permafrost peatland complexes"

_Hydrology and Earth System Sciences, 2020_

## Referee Comment (RC1) · Anonymous Referee #1 · 22 Oct 2020

This study needs more effort on the concept, the analysis, and the writing. My major critique on the **concept** is that the large scale analysis is not linked well with the conceptual model. What does the conceptual model mean at a larger scale? What is the timeframe when we would expect such changes? How big of an area is likely to change when? I am missing the space-for-time substitution that is mentioned in the abstract. This would improve the scientific significance. At the current stage, it is not clear what the new contribution of the study is.

Parts of the **analysis** itself are questionable, sometimes because they are just not well enough described. The statistical analysis with ANOVA cannot be used for autocorre-

lated data (such as the monthly values in this case); also comparisons should always be limited to the common period as with climate change most variables are certainly not stationary. Changes in permafrost area are mentioned in several places, but it is not clear how the permafrost area was estimated.

The **writing** needs to be more specific on what the authors did for the current study. In multiple places of the paper it is hard to distinguish between their work and other peoples work. The paper would be much easier to read if they used the active voice for everything they did and found out. It is also important that they separate the results from the discussion. That would help a lot to distinguish what the new contribution in this study is as compared to previous understanding and the literature cited. This is something that needs to be highlighted. In the current version, the joined section reads like a literature review in lots of paragraphs. Even the methods section includes parts that should be moved to the discussion or introduction. The description of the methods is, in many places, not clear and for some of the described methods it is not clear to me which results they generated. The complete methods section should be restructured (suggestion below) and the remote sensing methods should be illustrated with a figure. In several parts I am also missing information on why a specific method/dataset was used. The English is fine but the quality of the figures could be improved. The complete paper is much longer than it would need to be to address the objectives; it would be better is it was more concise.

Please also note the supplement to this comment:
https://hess.copernicus.org/preprints/hess-2020-411/hess-2020-411-RC1-supplement.pdf

**Supplement:**

**Review on**

**The trajectory of landcover change in peatland complexes with discontinuous permafrost, northwestern Canada**

Olivia Carpino, Kristine Haynes, Ryan Connon, James Craig, Élise Devoie and William Quinton

**Short summary**

The authors describe a conceptual model of landscape development from a permafrost underlain forest to a treeless wetland and, as last step, an afforested wetland. This conceptual model is underlain with historical and recent aerial photographs and energy and water balance data of one field site to describe the transformation in more detail. The study is motivated by a large scale analysis of the spatial distribution of landcover types in northwestern Canada.

**General comments**

This study needs more effort on the concept, the analysis, and the writing. My major critique on the **concept** is that the large scale analysis is not linked well with the conceptual model. What does the conceptual model mean at a larger scale? What is the timeframe when we would expect such changes? How big of an area is likely to change when? I am missing the space-for-time substitution that is mentioned in the abstract. This would improve the scientific significance. At the current stage, it is not clear what the new contribution of the study is.

Parts of the **analysis** itself are questionable, sometimes because they are just not well enough described. The statistical analysis with ANOVA cannot be used for autocorrelated data (such as the monthly values in this case); also comparisons should always be limited to the common period as with climate change most variables are certainly not stationary. Changes in permafrost area are mentioned in several places, but it is not clear how the permafrost area was estimated.

The **writing** needs to be more specific on what the authors did for the current study. In multiple places of the paper it is hard to distinguish between their work and other peoples work. The paper would be much easier to read if they used the active voice for everything they did and found out. It is also important that they separate the results from the discussion. That would help a lot to distinguish what the new contribution in this study is as compared to previous understanding and the literature cited. This is something that needs to be highlighted. In the current version, the joined section reads like a literature review in lots of paragraphs. Even the methods section includes parts that should be moved to the discussion or introduction. The description of the methods is, in many places, not clear and for some of the described methods it is not clear to me which results they generated. The complete methods section should be restructured (suggestion below) and the remote sensing methods should be illustrated with a figure. In several parts I am also missing information on why a specific method/dataset was used. The English

25 is fine but the quality of the figures could be improved. The complete paper is much longer than it would need to be to address the objectives; it would be better is it was more concise.

**Specific comments**

**Abstract** I waited until the last sentence to learn about the main method of this paper and I find it difficult to extract the main result and the main message from the abstract. I suggest to put the information about the method right after the first topic
30 introduction sentences, be more specific in the results sentences and add a statement about the implication of this work.

**18** 'This study explores the current trajectory of landcover change across...' this is what I would like to see in the study. However, the large scale analysis is quite disconnected from the rest.

**19** Where are you doing a space for time substitution? Mostly, you use a single location (Scotty Creek) as a substitution for a large area and show how it evolved over time.

35 **22–24** This needs to be included in Figure 3.

**Key points** The key points are all about methods. I suggest to include at least one on your findings.

**Introduction** The introduction touches on many interesting points. However, I have trouble to follow the introduction as the paragraphs do not seem to have one clear focus each and build on each other. Maybe you could slightly reorder the sentences and start every paragraph with a topic sentence, for example introducing current landscapes, observed changes,
40 implications for the water and energy budget. The current last paragraph is very helpful.

**40** Please update this reference to a peer-reviewed paper which also includes the thaw component.

**43** What is 'not well understood'? Two sentences later you write that lots of changes have already been documented. Please be more specific on the lack of knowledge.

**92–98** This part of the paragraph seems to belong to the second paragraph of the introduction.

45 **Figure 1** All fonts are too small, some are barely readable. Please use vector graphics (such as pdf) for all figures so that the resolution is high and the text does not appear blurry. What does the yellow line mean? Can you indicate with keywords what it separates? How was the border between the permafrost regions determined? Are you using the map by Brown et al.?

**Section 2.1.1** One or two pictures (maybe as part of Figure 1) would be good to show the different landscape parts. Please
50 indicate where permafrost can be found.

**167–169** Can you specify how much the temperature increased at this site specifically?

**179–181** 'relatively long record', 'Long-term observations': how many years? Are those continuous measurement series?

**Section 2.2** is not very clear to me. The methods could be described more clearly and I would like to read some sentences on why a certain wavelength/dataset/... was selected. The section would also profit a lot from a figure showing a small example area in all the different datasets and computed products. A table would also help, stating the most important properties for each dataset such as spatial coverage, resolution, date of acquisition, categories contained, who created it, citation. This could also be moved to the appendix.

**194** How is the warm season defined? How can you exclude moisture variations? I assume that you have different acquisition dates and some may be after a rainfall.

**189–198** Did I understand it correctly, that you selected one image per scene based on fewest possible clouds, latest possible year, and month in June/July/August? Maybe you could make the collection criteria more clear. Concerning vegetation development, the beginning of June is quite different as compared to mid of August. Can you justify the 'rendering the images seasonally comparable' a bit better?

**198–199** Why did you restrict yourself to those 3 bands? Can you explain why you did not include more?

**205–207** I do not know this dataset. Can you describe it briefly (What variables? Continuous or in classes? Spatial resolution? Vector or raster data?) Did you apply thresholds? Please cite a documentation and not only the download link.

**207–209** I also do not know this dataset and the reference does not appear in the literature list. Can you describe the dataset briefly (What variables? Continuous or in classes? Spatial resolution? Vector or raster data?) Did you apply thresholds? Please cite a documentation.

**203–211** This step seems central to me and it is very abstract. It would be great to see a figure with examples of the datasets in combination with the Landsat imagery and the result of your filtering. One small area in all the different images.

**212–214** It is not clear to me what you clustered here. Is it 3-band-Landsat pixels?

**215–217** How did you aggregate the classes? Manually based on expert knowledge? What was the advantage of having the unsupervised clustering first, if you targeted the specific classes described?

**220** Which map of peatland distribution?

**212–221** Can you add the clustering result to the figure I suggested?

**228–229** What do you mean by 'This generated a spatially distributed dataset'? I thought you reduced the spatial dimension to a north-south gradient.

**Section 2.3** is long and confusing. It should be restructured and some parts should be move to the introduction or discussion. It should contain only references to specific methods/datasets and not general findings on landscape change. I suggest to arrange the complete methods section as follows:

1. The Taiga Plains ecozone

2. The Scotty Creek field site

3. Geomatics methods

85 4. Water balance data (currently lines 234–237, 252–255, 260–282)

5. Imagery of the Scotty Creek basin (currently lines 283–295, 328–333)

6. Energy balance data (currently lines 299–323)

7. Conceptual model (currently lines 324–328, 333–340, needs more details)

**237–252** I do not see how these paragraphs fit into the section 'field based methods'. Please see my general comments.

90 **253** 'Interannual': please specify which years.

**255–259** This belongs to the discussion not to the methods.

**260–262** Please indicate how runoff was measured.

**Table 1** The caption sentence 'Both..thaw' should be moved elsewhere (The methods section on Scotty Creek would be a good place). You mention a runoff increase above. For non-stationary processes, it is misleading to calculate residuals
95 from variables averaged over different time periods. Please use the common period 2013-2016 to calculate residuals. The numbers from the complete datasets can be added as additional columns or rows.

**277–278** Table 1 does not show annual values. I think annual values would be interesting to see the variability and showing them would answer to my comment on Table 1. Why do you not show a figure with the time series of annual precipitation, runoff, evapotranspiration, and residual storage which you used for your study.

100 **280–282** Why do you analyse the runoff, evapotranspiration, and storage data for trends but not the precipitation data?

**293–295** By you or by the other authors you cite above?

**296 – 299** This does not belong in a methods section.

**318** Here you mention 'subcanopy' are all measurements below the canopy? Please specify this when you describe the stations.

**320–323** What do you test with the ANOVA? What are your responses and drivers? Are you looking for the effect of station
105 landcover on November reflected shortwave radiation, for example?

**328–333** What are these images used for as comparred to the airborne and satellite images described in l. 283–295?

**324–328, 333–340** Here you touch on the conceptual model you developed, but it is not clear to me how you did it. The methods section should describe how you did your analysis and why you used a specific method/dataset, but not why you study something in general (this part should be moved to the introduction or maybe partly to the discussion section).
110 You do not need to mention here which figures you show later.

**337–340** Whether or not your results can be extrapolated is a topic for the discussion section, not for the methods.

**Figure 2** What did you use the Landsat 8 data for in this map? As far as I understood, it was only used to estimate forest cover and not whether or not the landscape was 'peatland-dominated'. Please make this more clear in your methods! Some fonts are too small.

**344–345** Is this a finding, or a part of the original definition of a peatland which you used in the classification?

**Figure 3** Please exchange the word 'proportional' with 'fractional'. You are not really showing 'proportional permafrost area', but only the rough classes. Are these from the Brown permafrost map? Do you have more detailed information? If not, please change the caption. The fonts are too small and the whole figure too big for the content. I would be interested to see an additional line for fraction of peatlands.

**365** 'wetland features, including collapse scar bogs, are most prevalent' - This would be interesting to show in Figure 3

**Section 3.2** It is not clear to me what the new part in this study is as compared to previous understanding and the literature cited.

**396–401** Is this something you found out, or is it described in literature? Please cite one or more relevant articles and explain why you adopted/changed the phases.

**418** You mention, that the transition is very fast (40 years). Please discuss speed in a bit more detail. Are there other studies? It would be good to add a rough timeframe in your Figure 4. The work of Claire Treat may be relevant here, e.g. *Treat CC, Jones MC. Near-surface permafrost aggradation in Northern Hemisphere peatlands shows regional and global trends during the past 6000 years. The Holocene. 2018;28(6):998-1010. doi:10.1177/0959683617752858* (maybe other papers of her are even more interesting). She includes afforested peatlands in her work, but the timescales for forest recovery were more like 450 – 1500 years.

**419–423** What makes you think it is unlikely? I would like to see more discussion here.

**Figure 4** Fonts are much too small. Why does incoming shortwave radiation change? Is it measured below canopy? In this case please rename this variable. I do not understand why storage changes across the gradient. As I understand it, storage is not a flux (like runoff and evapotranspiration) and the storage change (which would be a flux as your other two variables) should be close to zero on a multiannual timescale. You do not mention the timeframe here.

**459–460** 'at the expense of permafrost' - how do you know? You do not describe soil temperature or ice content anywhere. Please be more specific when you talk about permafrost.

**Section 2.2.2** Please change your statistical analysis to incorporate autocorrelation and to use only the common period of all measurements. Please also do not provide exact p values but restrict yourself to $p < 0.05$ (or whatever threshold you use).

**519–520** Is this measured above or below canopy? Is there shading on the sensors? I do not see why (given the small distance) incoming radiation should be different. I suggest to show albedo instead. It is more interesting and has more implications on the energy partitioning within the vegetation. This comment of course also applies at your statistical analysis and Figure 5.

**529–531** It is not statistically sound to use ANOVA on a timeseries (of monthly values in this case). The reason is, that the values are highly autocorrelated. Therefore, you get a 'fake confidence' and the p values are wrong. Either (I) remove your statistical analysis including all p values, (II) use an appropriate methods to include autocorrelation, or (III) use data with no (or at least little) autocorrelation, such as annual values. You could also analyse all mean June values in one analysis, as June 1999 should not be correlated to June 2000. This would give you one p value per month.

**Figure 5** As described, your p values are wrong. However, if you fix the analysis, please anyway only write $p<0.05$ (or whatever threshold you use), not p=.... This gives false confidence. Please consider changing to $W/m^2$, which is used more often than $MJ/m^2/day$. Please use only the common measurement period as the fluxes are likely not stationary. Fonts too small (use the caption font as an indicator of an appropriate size). Did you check what the low values of longwave radiation represent ($10$–$15\,MJ/m^2/day$)? Are these real observations or issues with the instruments?

**545–563** Given the small distance between the sites, incoming radiation is not very interesting. Please analyse albedo instead. It would also be interesting to estimate snowmelt timing at the different sites and analyse that. In particular as you discuss albedo later.

**Section 3.2.3** Do you have new numbers or results to add to this literature review? What do you mean by 'runoff', only in streams/rivers or also as groundwater?

**617–620** How about mosses?

**666–667** Does landcover depend on climate or climate on landcover?

**667–669** Did you really show how the changes were initiated?

**675–678** Where exactly can I find these results?

**681–684** Here again you mention that your research was on permafrost distribution. However, you did not analyse permafrost distribution. If you prefer to keep the statements on permafrost, you need to make it more clear how you measure or estimate permafrost distribution.

**Specific comments**

**\*** It would have been more convenient if you used hyperlinks so I could click on the references.

**183** 'which are not only found extensively': the studies or the collapse scars? Maybe rephrase the sentence.

**221** Start a new paragraph for this new thought.

170 **238** describe**s**

**237–244** These two sentences are very long and complicated. Please split them into more sentences.

**251** With 'this study', do you mean your current paper, or Quinton 2019?

**252–253** Sentence a bit confusing. How about 'We used precipitation data of the years ... to ... collected by the SCRS.'

**569** The word 'plotted' here and at lot of places in the manuscript irritates me. Try to omit it.

---

## Referee Comment (RC2) · Anonymous Referee #2 · 7 Dec 2020

The manuscript entitled "The trajectory of landcover change in peatland complexes with discontinuous permafrost, northwestern Canada" by Olivia Carpino et al. focuses on the Taiga Plains of Northwest Canada, where rapid climate warming has significantly reduced the area underlain by permafrost in peatland complexes. A massive landscape shift has occurred in recent years, from a forest-dominated landscape to a wetland-dominated landscape. The authors explore the current trajectory of land cover change in a 300,000 km^2 area of discontinuous permafrost in northwestern Canada by presenting spatiotemporal variability using a 600-km latitudinal span of this region. By combining extensive geomatics data with ground-based meteorological and hydrological measurements, a new conceptual model of landscape evolution was developed.

[Figure]

This model explains the observed patterns of land cover change caused by permafrost thawing and provides a basis for predicting future changes. This is a very interesting paper and provides deep insights into how future permafrost loss may change the Taiga forest. The conceptual model and the discussion of water and energy balances are also interesting. However, I feel that the paper is somewhat disorganized and needs a more unified structure. I also think the conceptual model needs to be validated, and I would recommend that the validation be described in terms of the results and discussion. Overall, the authors need to revise the manuscript before its publication. Although there are some issues, I recommend that this paper be published after revisions are made.

Specific comments (1) Sections 3.1 and 3.2 of the Results and Discussion in Chapter 3 are in a completely different vein, making it difficult to read. Section 3.1 discusses the latitudinal distribution of forest cover and permafrost. I feel it would be better to show the percentage of peat areas and wetlands along with latitude in Figure 3. Similarly, the spatial distribution of forests and wetlands can be shown in a figure similar to Figure 2. Also, the spatial distribution is clearly shown in Figures 2 and 3, but the valuable aerial photos and data of IKONOS are described in the method of Chapter 2, which have been analyzed since 1947. I would like to see a figure similar to Figures 2 and 3, one that shows the changes over time based on the data analysis.

(2) At the beginning of Section 3.2, a conceptual model of landscape change associated with the thawing of frozen ground is provided in Figure 4. The conceptual model is introduced so abruptly that it feels as though it has not been validated. Therefore, I would like to see the conceptual model validated on the basis of the analytical data in section 3.1. I would like you to show the results of the verification of the proposed model on the whole study region, using the data from Section 3.1, although it is valid for Scotty Creek. In addition, the purpose of this study was to characterize end members and intervening stages of the landcover transition. Please indicate the end members in Figure 4.

(3) There is no single designation for Scotty Creek; please clarify if Scotty Creek Research Station (SCRS) is the same as Scotty Creek, Scotty Creek basin or Scotty Creek watershed. I recommend that Scotty Creek be unified with SCRS or others. In addition, I feel that there needs to be a map of the meteorological and hydrological observations that are being made at SCRS. In particular, a description or table of the four component radiation observations is needed. In Figure 5, it is difficult to understand the changes in the four radiative components without a description of whether they are observations above the vegetation canopy or on the forest floor. Additionally, please show which stage of the conceptual model each letter (a)-(d) corresponds to.ãĂĂ

(4) Lines 106-113: Here the characteristics of the energy balance of the forest canopy are described. However, the difference between wetlands and forests is also evident in the water balance. For example, the amount of precipitation reaching the ground due to rainfall and snowfall interception is smaller in forests, making them more prone to drying out than in wetlands. I think it is important to describe this point as well.

(5) Line 135: Dry peat has been mentioned, but I think it is necessary to mention rainfall interception by mosses and other factors as a cause (e.g., Price et al., 1997, J. Hydrol, Suzuki et al., 2007, HYP). The reason for the dryness of the peat layer is that mosses are thought to play a major role in blocking rainfall. It would be useful to describe the underlying vegetation of the forest, and such descriptions should be added.

---

## Author Comment (AC1) · 12 Feb 2021

REVIEWER #1

Review on The trajectory of landcover change in peatland complexes with discontinuous permafrost, northwestern Canada Olivia Carpino, Kristine Haynes, Ryan Connon, James Craig, Élise Devoie and William Quinton

Short summary The authors describe a conceptual model of landscape development from a permafrost underlain forest to a treeless wetland and, as last step, an afforested wetland. This conceptual model is underlain with historical and recent aerial photographs and energy and water balance data of one field site to describe the transformation in more detail. The study is motivated by a large scale analysis of the spatial distribution of landcover types in northwestern Canada.

General comments This study needs more effort on the concept, the analysis, and the writing. My major critique on the concept is that the large scale analysis is not linked well with the conceptual model. What does the conceptual model mean at a larger scale? What is the timeframe when we would expect such changes? How big of an area is likely to change when? I am missing the space-for-time substitution that is mentioned in the abstract. This would improve the scientific significance. At the current stage, it is not clear what the new contribution of the study is.

Parts of the analysis itself are questionable, sometimes because they are just not well enough described. The statistical analysis with ANOVA cannot be used for autocorrelated data (such as the monthly values in this case); also comparisons should always be limited to the common period as with climate change most variables are certainly not stationary. Changes in permafrost area are mentioned in several places, but it is not clear how the permafrost area was estimated.

The writing needs to be more specific on what the authors did for the current study. In multiple places of the paper it is hard to distinguish between their work and other peoples work. The paper would be much easier to read if they used the active voice for everything they did and found out. It is also important that they separate the results from the discussion. That would help a lot to distinguish what the new contribution in this study is as compared to previous understanding and the literature cited. This is something that needs to be highlighted. In the current version, the joined section reads like a literature review in lots of paragraphs. Even the methods section includes parts that should be moved to the discussion or introduction. The description of the methods is, in many places, not clear and for some of the described methods it is not clear to me which results they generated. The complete methods section should be restructured (suggestion below) and the remote sensing methods should be illustrated with a figure.

In several parts I am also missing information on why a specific method/dataset was used. The English is fine but the quality of the figures could be improved. The complete paper is much longer than it would need to be to address the objectives; it would be better is it was more concise.

Response (R): We thank the reviewer for these suggestions. We appreciate the detailed and thorough review. The introduction section will be completely rewritten and the methods section will be largely rewritten and restructured to address some of these concerns. These changes will provide more context for the study and will improve the linkages between the conceptual framework (which will not be referred to as a conceptual model in response to this comment and others) and analysis. We will also clarify the new contributions of this study.

Specific comments Abstract I waited until the last sentence to learn about the main method of this paper and I find it difficult to extract the main result and the main message from the abstract. I suggest to put the information about the method right after the first topic introduction sentences, be more specific in the results sentences and add a statement about the implication of this work.

R: As suggested, the abstract will be revised and restructured in order to first introduce the topic of the paper, identify the methods used and present some of the key findings of the work. We will clarify the major results of the research and, through the restructuring of the abstract, clarify its overall message.

18 'This study explores the current trajectory of landcover change across...' this is what I would like to see in the study. However, the large scale analysis is quite disconnected from the rest.

R: We will clarify our approach in the abstract by expanding on the sentence identified by the reviewer here. We will also provide more detail as to how the broader geomatics data spanning the Taiga Plains is integrated with our field-based hydrometeorological data collected in the Scotty Creek basin, located within the Taiga Plains. The large-

scale change in forested peatland cover observed across the latitudinal, and therefore climatic, gradient of the Taiga Plains is reflective of observations of localized change from permafrost plateau to collapse scar wetland and ultimately afforested wetland occurring in the Scotty Creek basin.

19 Where are you doing a space for time substitution? Mostly, you use a single location (Scotty Creek) as a substitution for a large area and show how it evolved over time.

R: The 600 km north-south latitudinal gradient of the Taiga Plains also represents a gradient in prevailing climate. Therefore, the broad-scale change in forested peatland cover across this climatic gradient is representative of the anticipated landscape changes in a localized area (represented by the Scotty Creek basin) within that gradient over time. The forested plateaus in the northern Taiga Plains and the afforested wetlands at the southern region of the Taiga Plains frame the endmembers of landscape transition. These endmembers as well as plateau-wetland complexes are all located within the Scotty Creek basin. Therefore, our knowledge of the hydrological and thermal mechanisms governing these landcover types from field-based research in the Scotty Creek basin provide us with an understanding of the mechanisms contributing to landscape change. These results can then be integrated with the geomatics results across the Taiga Plains to anticipate the trajectory of change in the discontinuous permafrost zone. We will further clarify this connection and the space-for-time substitution both in the abstract and in the Methods section of the revised manuscript.

22–24 This needs to be included in Figure 3.

R: To clarify the concurrent changes in wetland coverage, labels will be added to the figure detailing the dominant wetland type occurring with the changing forested area with latitude. At the high latitudes, "Forest with permafrost" will be added, while "Collapse Scars" and "Permafrost-free Forest" will be added at the mid- and low-latitudes, respectively.

Key points The key points are all about methods. I suggest to include at least one on

your findings.

R: We will edit the Key Points to mention our methodological approach in one point as well as include two points on our findings. Key Points will read as follows in the revised manuscript: 1. Conceptual framework developed to understand the trajectory of permafrost thaw-induced land cover change 2. Permafrost thaw-induced land cover changes vary latitudinally across the plateau-wetland complexes of the discontinuous permafrost zone 3. Partial wetland drainage triggers ecohydrological and thermal feedbacks that promote reforestation after full permafrost thaw

Introduction The introduction touches on many interesting points. However, I have trouble to follow the introduction as the paragraphs do not seem to have one clear focus each and build on each other. Maybe you could slightly reorder the sentences and start every paragraph with a topic sentence, for example introducing current landscapes, observed changes, implications for the water and energy budget. The current last paragraph is very helpful.

R: We thank the reviewer for this comment. The Introduction will be entirely re-written to improve the focus.

40 Please update this reference to a peer-reviewed paper which also includes the thaw component.

R: The following peer-reviewed reference will be added to replace the original reference: "Box, J.E., Colgan, W.T., Christensen, T.R., Schmidt, N.M, Lund, M. . . . Olsen, M.S. (2019). Key indicators of Arctic climate change: 1971-2017. Environmental Research Letters, 14, 045010. https://doi.org/10.1088/1748-9326/aafc1b"

43 What is 'not well understood'? Two sentences later you write that lots of changes have already been documented. Please be more specific on the lack of knowledge.

R: The introduction will be entirely rewritten to improve clarity.

92–98 This part of the paragraph seems to belong to the second paragraph of the

introduction.

R: As requested, we will move these lines of text to the end of the second paragraph of the Introduction.

Figure 1 All fonts are too small, some are barely readable. Please use vector graphics (such as pdf) for all figures so that the resolution is high and the text does not appear blurry. What does the yellow line mean? Can you indicate with keywords what it separates? How was the border between the permafrost regions determined? Are you using the map by Brown et al.?

R: Font sizes will be increased and the figure will be exported at a higher dpi resolution to improve the clarity of the image. We will add the label of "Taiga Plains ecoregion" with an arrow pointing to the yellow line. We will add a sentence to the figure caption that the boundary between the sporadic and continuous is from Brown et al. 2002.

Section 2.1.1 One or two pictures (maybe as part of Figure 1) would be good to show the different landscape parts. Please indicate where permafrost can be found.

R: An additional figure will be added to show the different land covers described.

167–169 Can you specify how much the temperature increased at this site specifically?

R: To clarify this point, the following text will be added: "Data collected by Environment and Climate Change Canada at the Fort Simpson A climate station show that MAAT has increased by approximately 0.05°C/year since 1950, with warming most pronounced during the winter."

179–181 'relatively long record', 'Long-term observations': how many years? Are those continuous measurement series?

R: This section will be rewritten. The length and nature of the Scotty Creek observations will be clarified with the following revised sentences: "Scotty Creek (61.3°N, 121.3°W) has been the focus of field studies and monitoring since the mid-1990s and

as such, the long-term and detailed data archive at Scotty Creek (Haynes et al., 2019) provide a unique opportunity to evaluate land cover changes over a period that coincides with rapid climate warming. As such, Scotty Creek also provides a reference to interpret land cover changes for terrains of the same type throughout the region." "The long-term monitoring in the Scotty Creek basin presents a unique opportunity to study warming-induced landcover changes to plateau-wetland complexes in the Taiga Plains given the record of field and modelling studies at this site over the course of nearly three decades. This record has coincided with a period of drastic climate warming. Long-term field research in the Scotty Creek basin, including continuous hydrometeorological observations in concert with annual and seasonal monitoring, facilitate the examination of the impacts of climate change on peat plateau-collapse scar wetland complexes. This landform type is found extensively both throughout northwestern Canada and across the global subarctic (Olefeldt et al. 2016)."

Section 2.2 is not very clear to me. The methods could be described more clearly and I would like to read some sentences on why a certain wavelength/dataset/... was selected. The section would also profit a lot from a figure showing a small example area in all the different datasets and computed products. A table would also help, stating the most important properties for each dataset such as spatial coverage, resolution, date of acquisition, categories contained, who created it, citation. This could also be moved to the appendix.

R: Much of this section will be rewritten and reorganized in order to better clarify the methods described in this manuscript. A figure will also be made to illustrate a sample area of each dataset. Further information on each dataset will be added in-text including information on spatial coverage, resolution, data categories, etc.

194 How is the warm season defined? How can you exclude moisture variations? I assume that you have different acquisition dates and some may be after a rainfall.

R: Yes, the 70 Landsat scenes used in this mosaic have varying acquisition dates. Soil

moisture variations were not impactful on classifying forest cover, which was the purpose of the Landsat mosaic. The Landsat methods will be clarified in the manuscript. The sentence referencing soil moisture variations will be removed to avoid confusion. The warm season is defined as the snow-free period. This will also be clarified in the manuscript by replacing "warm season" with "snow-free season".

189–198 Did I understand it correctly, that you selected one image per scene based on fewest possible clouds, latest possible year, and month in June/July/August? Maybe you could make the collection criteria more clear. Concerning vegetation development, the beginning of June is quite different as compared to mid of August. Can you justify the 'rendering the images seasonally comparable' a bit better?

R: Yes, that is correct. As the main purpose of the Landsat imagery was to identify black spruce-dominated forests, the absence of snow was the main priority. Other vegetation development is much more variable between June and August. The sentences beginning at line 194 in the original manuscript were rewritten to read: "Acquiring imagery during the snow-free season was prioritized and as such, all 70 Landsat tiles were acquired in June, July, or August, rendering the coniferous forest cover seasonally comparable and allowing for a more streamlined mosaicking process."

198–199 Why did you restrict yourself to those 3 bands? Can you explain why you did not include more?

R: The Landsat data was used for the purpose of classifying forest cover within identified peat plateau wetland complexes. As such the three bands used were deemed sufficient. Additional bands may have been useful had the Landsat data been used in another way. Near infrared (red), red (green), green (blue) (Landsat 8 bands 5, 4, 3) is a traditional band combination useful for visualizing vegetation characteristics.

205–207 I do not know this dataset. Can you describe it briefly (What variables? Continuous or in classes? Spatial resolution? Vector or raster data?) Did you apply thresholds? Please cite a documentation and not only the download link.

R: The following sentences will be added to the manuscript to provide further details: "The saturated soils dataset is part of a larger digital cartographical project by Natural Resources Canada, CanVec. The CanVec dataset is a vector format dataset, which can be downloaded by province/territory or Canada-wide and includes over 60 features organized into 8 themes, including land features. Land features, including the distribution of saturated soils, were digitized at a scale of 1:50000 (Natural Resources Canada 2017)."

207–209 I also do not know this dataset and the reference does not appear in the literature list. Can you describe the dataset briefly (What variables? Continuous or in classes? Spatial resolution? Vector or raster data?) Did you apply thresholds? Please cite a documentation.

R: We thank the reviewer for bringing this missing reference to our attention. The reference to this dataset will be added. Two additional documentation references will also be added. The following sentences will be added to the manuscript to provide further information on this dataset: "The NCSCD is also a polygon database developed by the Bolin Centre for Climate Research through synthesizing data from numerous regional and national soil maps alongside field-data collected across Canada, USA, Russia, and the EU. The NCSCD includes data on the fractional coverage of different soil types and stored soil organic carbon (Hugelius et al. 2013a; Hugelius et al. 2013b). While the original format of the NCSCD is a vector, gridded data is also available at resolutions varying from 0.012 to 1 for circum-arctic use (Hugelius et al. 2013b). In addition to the circum-arctic dataset, the NCSCD is also available on a country-wide or regional scale, including a Canada product (Hugelius et al. 2013b)." The added references are as follows: Bolin Centre for Climate Research (2013). The Northern Circumpolar Soil Carbon Database. Available at: https://bolin.su.se/data/ncscd/ (Accessed March 20, 2019) Hugelius G., Bockheim J.G., Camill P., Elberling B., Grosse G., Harden J.W., Johnson K., Jorgenson T., Koven C.D., Kuhry P., Michaelson G., Mishra U., Palmtag J., Ping C.-L., O'Donnell J., Schirrmeister L., Schuur E.A.G., Sheng Y., Smith L.C., Strauss

J. and Yu Z. (2013a). Earth System Science Data, 5, 393–402. DOI:10.5194/essd-5-393-2013 Hugelius, G., Tarnocai, C., Broll, G., Canadell, J. G., Kuhry, P., and Swanson, D. K. (2013b) Earth System Science Data, 5, 3–13. DOI:10.5194/essd-5-3-2013

203–211 This step seems central to me and it is very abstract. It would be great to see a figure with examples of the datasets in combination with the Landsat imagery and the result of your filtering. One small area in all the different images.

R: The new methods figure mentioned above will also include this.

212–214 It is not clear to me what you clustered here. Is it 3-band-Landsat pixels?

R: The unsupervised Iso Cluster classification was done on the Landsat mosaic.

215–217 How did you aggregate the classes? Manually based on expert knowledge? What was the advantage of having the unsupervised clustering first, if you targeted the specific classes described?

R: Yes, the aggregation was completed manually. The unsupervised approach is an effective approach due to the large latitudinal span as well as the classification being completed on areas already identified as likely peat plateau-wetland complexes. Unsupervised classifications have the benefit of maximizing the number of classes, which can be especially useful if unexpected or uncommon classes are found. Manual aggregation is very commonly performed on unsupervised classifications. When using manual aggregation, first-hand knowledge of the area or imagery can then be used to aggregate the spectral classes into classes that are more meaningful to the specific study. Further text will be added throughout the geomatics methods to clarify the approaches used.

220 Which map of peatland distribution?

R: The map of peatland distribution referenced is the peatland distribution identified in the previous paragraph, which was developed using the saturated soils dataset from NRCan and the NCSCD. This will be clarified by introducing this earlier in the same
paragraph.

212–221 Can you add the clustering result to the figure I suggested?

R: A new figure (as mentioned above) will be added to the geomatics section.

228–229 What do you mean by 'This generated a spatially distributed dataset'? I thought you reduced the spatial dimension to a north-south gradient.

R: That is correct. The words "spatially distributed" will be removed.

Section 2.3 is long and confusing. It should be restructured and some parts should be move to the introduction or discussion. It should contain only references to specific methods/datasets and not general findings on landscape change. I suggest to arrange the complete methods section as follows: 1. The Taiga Plains ecozone 2. The Scotty Creek field site 3. Geomatics methods 4. Water balance data (currently lines 234–237, 252–255, 260–282) 5. Imagery of the Scotty Creek basin (currently lines 283–295, 328–333) 6. Energy balance data (currently lines 299–323) 7. Conceptual model (currently lines 324–328, 333–340, needs more details)

R: To improve the clarity of the descriptions of the study sites and methods, these sections will be re-organized as suggested. References to findings of landscape change that are not critical to the explanation and justification of the methods used in this study will be removed from the Methods section. Overall, the Study Sites and Methods sections will be restructured with the following headings in order to clarify our methodological approach: 2. Study Site 2.1 The Taiga Plains Ecozone 2.2 Scotty Creek, Northwest Territories 3. Methods 3.1 Geomatics Methods 3.2 Scotty Creek Imagery 3.3 Hydrological Data 3.4 Radiation Fluxes The hydrological, radiation and land cover data used in the development of the conceptual framework (presented in the Results/Discussion sections) are each described in the appropriate Methods sub-section above. The conceptual model will be left to the Results/Discussion section.

237–252 I do not see how these paragraphs fit into the section 'field based methods'.

Please see my general comments.

R: With these sentences, we are aiming to explain the relationship between the broad-scale landscape change occurring in the plateau-wetland complexes of the Taiga Plains and the localized permafrost thaw-induced changes observed within the Scotty Creek basin. With the restructuring of the Methods section, this explanation will no longer be associated with the "field-based methods". Instead, the "field-based methods" section has been further divided into section 3.2, 3.3, 3.4 as noted above. Additionally, we will edit these statements to clarify the connection between the large-scale trends and our observations of localized landcover and hydrometeorological change in the Scotty Creek basin.

253 'Interannual': please specify which years.

R: The word 'interannual' will be removed and the precipitation monitoring years of 2008 to 2019 will be specified.

255–259 This belongs to the discussion not to the methods.

R: These sentences will be removed from the Methods and the concept of increasing hydrological connectivity (as opposed to precipitation inputs) contributing to observed elevated runoff from the Scotty Creek basin will be mentioned in the Discussion where appropriate.

260–262 Please indicate how runoff was measured.

R: Discharge from Scotty Creek has been measured by the Water Survey of Canada since 1995. The gauged portion of the basin spans 152 km2. This information will be provided in the revised manuscript.

Table 1 The caption sentence 'Both..thaw' should be moved elsewhere (The methods section on Scotty Creek would be a good place). You mention a runoff increase above. For non-stationary processes, it is misleading to calculate residuals from variables averaged over different time periods. Please use the common period 2013-2016 to calculate residuals. The numbers from the complete datasets can be added as additional columns or rows.

R: This sentence will be removed from the Table 1 caption and incorporated into the overarching explanation of the development of the conceptual framework in the rewritten Methods sections. The dynamic landscape of the Scotty Creek basin is changing over time as permafrost thaws as a result of climate warming. Therefore, the runoff time series essentially represents different transitional stages with different proportions of each land cover type comprising the overall basin landscape. Consequently, we are selecting representative runoff values associated with the different stages of landscape change (permafrost-dominated forest as compared to wetland-dominated and subsequently permafrost-free forest). We will highlight this in the revised manuscript so as to clarify the reasoning for not utilizing the common time period of 2013-2015.

277–278 Table 1 does not show annual values. I think annual values would be interesting to see the variability and showing them would answer to my comment on Table 1. Why do you not show a figure with the time series of annual precipitation, runoff, evapotranspiration, and residual storage which you used for your study.

R: Table 1 shows annual values. Figure 4a (in the results/discussion) shows how these components change over time, though precipitation is not included as no changes in annual precipitation have been observed. The caption to Figure 4a will be modified to indicate the connection to Table 1.

280–282 Why do you analyse the runoff, evapotranspiration, and storage data for trends but not the precipitation data?

R: Previous research in the Scotty Creek basin has observed virtually no change in total annual precipitation throughout the period of data collection (Connon et al. 2014; Haynes et al. 2018). Therefore, we chose not to display the precipitation trends in the water balance portion of the conceptual framework, opting rather to display only runoff, storage and evapotranspiration, which are hydrological indicators of landscape change

in this environment. However, annual precipitation values are used in the calculation of the storage term as a residual. We will clarify the reasoning for not incorporating precipitation visually into the water balance of the conceptual framework in sub-section "3.3 Hydrological Data" of the revised manuscript.   293–295 By you or by the other authors you cite above?

R: The quantification of landcover changes in this study incorporates the aerial photo and remote sensing imagery classifications originally presented by Quinton et al. (2010), Carpino et al. (2018) and Disher (2020). However, our work takes these previous studies to present a comprehensive examination of landscape change in the Scotty Creek basin over the complete period of the imagery record (1947 to 2018). This will be clarified in the sub-section "3.2 Scotty Creek Imagery" of the revised manuscript.

296 – 299 This does not belong in a methods section.

R: This statement will be removed from the Methods section in the revised manuscript.

318 Here you mention 'subcanopy' are all measurements below the canopy? Please specify this when you describe the stations.

R: The following sentence will be added to the first paragraph of section 3.4 to clarify radiation measurements: "All radiation measurements were made below the tree canopy at a height of 2 m above the ground surface."

320–323 What do you test with the ANOVA? What are your responses and drivers? Are you looking for the effect of station land cover on November reflected shortwave radiation, for example?

R: The ANOVA tested whether or not there was a significant effect of landcover (driving variable) on monthly shortwave and longwave incoming and outgoing radiation (response variables). Changes in each of the energy components over time are not statistically tested in this study.

328–333 What are these images used for as compared to the airborne and satellite

images described in l. 283–295?

R: The RPAS imagery is used to visually illustrate the stages of landscape change identified in our conceptual framework. These aerial images, all collected in the Scotty Creek basin, illustrate with the use of photographs the mosaic-like landscape in this region. In contrast, the aerial photos and satellite imagery are used to quantify the proportions of the landscape represented by each of the peat plateau, collapse scar and afforested wetland landcover types in the Scotty Creek basin over the period of available imagery (1947 to 2018). We will clarify the distinction in the application of the RPAS imagery and aerial photos and satellite imagery in the sub-section "3.2 Scotty Creek Aerial Imagery" of the revised manuscript.

324–328, 333–340 Here you touch on the conceptual model you developed, but it is not clear to me how you did it. The methods section should describe how you did your analysis and why you used a specific method/dataset, but not why you study something in general (this part should be moved to the introduction or maybe partly to the discussion section). You do not need to mention here which figures you show later.

R: In the new sub-sections pertaining to the water balance, energy balance and aerial imagery field data collection, we will remove references to the overarching motivation and concepts that provide the foundation of the conceptual framework. With the restructuring of the Methods section as explained above, we will clarify the development of the conceptual framework in the rewritten methods section. Additionally, in the associated sub-sections, we will clarify how the hydrology, energy and RPAS imagery components are incorporated into the water balance, energy balance and quantification of landcover change in the conceptual framework.

337–340 Whether or not your results can be extrapolated is a topic for the discussion section, not for the methods.

R: The purpose of the text in lines 337-340 is to explain the approach that we have taken of using ground-based observation at an intensively studied site (i.e. Scotty

Creek) to interpret remotely sense ground surface changes over a latitudinal transect. The new text will read as follows: "Therefore, ongoing shifts along the proposed trajectory of change in landcover and the associated hydrometeorological changes in Scotty Creek can be extrapolated to other similar peat plateau-collapse scar wetland sites, both throughout the Taiga Plains and the global subarctic."

Figure 2 What did you use the Landsat 8 data for in this map? As far as I understood, it was only used to estimate forest cover and not whether or not the landscape was 'peatland-dominated'. Please make this more clear in your methods! Some fonts are too small.

R: We thank the reviewer for noticing that Landsat 8 data was included in the caption, this will be removed. Landsat 8 data was only used to estimate forest cover rather than predict the distribution of peatland-dominated terrain. Fonts will be increased on figures throughout the manuscript.

344–345 Is this a finding, or a part of the original definition of a peatland which you used in the classification?

R: This was a finding of the present study. The methods used are summarised in the caption for Figure 2. To clarify this point, the opening sentence of section 3.1 will be changed to the following: "To place the analysis for Scotty Creek described above into a regional context, geomatics methods were applied to both zones of discontinuous permafrost within the Taiga Plains to quantify the areas occupied by each of the major land covers of all areas identified as peatland-dominated lowland."

Figure 3 Please exchange the word 'proportional' with 'fractional'. You are not really showing 'proportional permafrost area', but only the rough classes. Are these from the Brown permafrost map? Do you have more detailed information? If not, please change the caption. The fonts are too small and the whole figure too big for the content. I would be interested to see an additional line for fraction of peatlands.

R: "Proportional" will be replaced with "Fractional" and the fonts will be increased. As mentioned above, the dominant wetland type coincident with the changing forest cover at the high, mid- and low latitudes will be annotated in this figure. The referenced Brown permafrost map was used, this information will be added to the figure caption.

365 'wetland features, including collapse scar bogs, are most prevalent' - This would be interesting to show in Figure 3

R: To clarify the concurrent changes in wetland coverage, labels will be added to the figure detailing the dominant wetland type occurring with the changing forested area with latitude. At the high latitudes, "Forest with permafrost" will be added, while "Collapse Scars" and "Permafrost-free Forest" will be added at the mid- and low-latitudes, respectively.

Section 3.2 It is not clear to me what the new part in this study is as compared to previous understanding and the literature cited.

R: The Introduction will be re-written, and new preamble will be added before the conceptual framework is presented. This new text will clarify what is new and what is synthesised from existing work.

396–401 Is this something you found out, or is it described in literature? Please cite one or more relevant articles and explain why you adopted/changed the phases.

R: These stages of land cover change form the basis of our proposed conceptual framework for permafrost thaw-driven land cover change in peatland dominated regions of thawing permafrost. This will be explained much more clearly as a result of the revised Introduction and new text preceding the presentation of this new framework.

418 You mention, that the transition is very fast (40 years). Please discuss speed in a bit more detail. Are there other studies? It would be good to add a rough timeframe in your Figure 4. The work of Claire Treat may be relevant here, e.g. Treat CC, Jones MC. Near-surface permafrost aggradation in Northern Hemisphere peatlands shows

regional and global trends during the past 6000 years. The Holocene. 2018;28(6):998-1010. doi:10.1177/0959683617752858 (maybe other papers of her are even more interesting). She includes afforested peatlands in her work, but the timescales for forest recovery were more like 450 – 1500 years.

R: Unlike traditional concepts of land cover change in peatland dominated regions of discontinuous permafrost in which forest re-establishment occurs over several centuries and is constrained by the rate of permafrost re-development, the concept presented here describes forest re-establishment as resulting from continued permafrost thaw, a process which removes "permafrost dams" and allows wetlands to de-water sufficiently for tree growth. This process occurs much more rapidly (within half a century) since the development of surface conditions that are sufficiently dry to support trees occurs much more rapidly as a result of wetland de-watering rather than as a result of permafrost re-establishment. New text explaining this difference will be added to the Introduction, and to the Results and Discussion section. Claire Treat's work will also be referenced in the context of this study, we thank the reviewer for this suggestion.

419–423 What makes you think it is unlikely? I would like to see more discussion here.

R: This sentence will be removed.

Figure 4 Fonts are much too small. Why does incoming shortwave radiation change? Is it measured below canopy? In this case please rename this variable. I do not understand why storage changes across the gradient. As I understand it, storage is not a flux (like runoff and evapotranspiration) and the storage change (which would be a flux as your other two variables) should be close to zero on a multiannual timescale. You do not mention the timeframe here.

R: The font size will be increased. Incoming radiation is measured below the canopy. We will add new text to the Methods section and to the Results and Discussion sections so that this is now clear. Insolation is therefore greatest over wetlands because they are treeless, and least on plateaus with a dense canopy cover. It will be explained

that as permafrost thaws, the density of the overlying canopy decreases and as a result, the insolation transitions toward the value measured over wetlands. From year to year the change in storage of the landscape is indeed near zero, however as we are discussing landscape evolution, the formation of talik features and collapse scars allow for more water to be retained on the landscape. As thaw progresses, more of the landscape becomes connected to the drainage network and contributes to runoff, reducing the overall storage capacity of the landscape. In this sense, the maximum storage capacity of the landscape changes as permafrost thaws, and this is what is meant by changing storage across the gradient. This will be clarified in the text.

459–460 'at the expense of permafrost' - how do you know? You do not describe soil temperature or ice content anywhere. Please be more specific when you talk about permafrost.

R: The revised introduction section will clarify this point by drawing on the large number of studies that demonstrate that permafrost thaw in the study region results in a transfer of forest (i.e. peat plateau) to treeless wetland. We will add more text to explain this permafrost thaw induced land cover change from permafrost (forest) to permafrost-free (wetland). The revised "Scotty Creek, Northwest Territories" section will contain information on permafrost characterization (soil temperature and composition, ice content, active layer thickness).

Section 2.2.2 Please change your statistical analysis to incorporate autocorrelation and to use only the common period of all measurements. Please also do not provide exact p values but restrict yourself to p<0.05 (or whatever threshold you use).

R: Changes in radiation components over time are not being considered with our statistical approach. Rather, the one-way ANOVAs were performed to assess potential differences in each of the four radiation components across the different land cover types examined in this study. The data from each meteorological station is never treated against itself, but is instead compared with the data from each of the other three stations. Changes in each of the energy components over time are not statistically tested in this study. We are testing for differences in each of the radiation components individually between the four land cover types. In the revised manuscript, p values will be expressed in relation to an alpha of 0.05.

519–520 Is this measured above or below canopy? Is there shading on the sensors? I do not see why (given the small distance) incoming radiation should be different. I suggest to show albedo instead. It is more interesting and has more implications on the energy partitioning within the vegetation. This comment of course also applies at your statistical analysis and Figure 5.

R: The revised Methods section will explain that the measurements were made below the canopy. Because of different canopy densities (sparse to dense) and because of adjacent forested (plateau) and treeless (wetland) terrains, insolation can vary widely over short distances. On the plateaus, albedo varies by less than 5%. The difference between plateaus and wetlands in terms of albedo is greater, but this difference is still small in comparison with the contrast in insolation between these two sites. The greatest contrast occurs in late winter while the snow cover is still present in the forests but absent in the adjacent wetlands. These points will all be incorporated and expanded upon in the revised manuscript.

529–531 It is not statistically sound to use ANOVA on a time series (of monthly values in this case). The reason is, that the values are highly auto-correlated. Therefore, you get a 'fake confidence' and the p values are wrong. Either (I) remove your statistical analysis including all p values, (II) use an appropriate methods to include autocorrelation, or (III) use data with no (or at least little) autocorrelation, such as annual values. You could also analyse all mean June values in one analysis, as June 1999 should not be correlated to June 2000. This would give you one p value per month.

R: Changes in radiation components over time are not being considered with our statistical approach. Rather, the one-way ANOVAs were performed to assess potential

differences in each of the four radiation components across the different land cover types examined in this study. Changes in each of the energy components over time are not statistically tested in this study. With respect to the issue of potential autocorrelation, the data from each meteorological station is never evaluated against itself, but is instead compared with the data from each of the other three stations. Therefore, as we are testing for differences in each of the radiation components individually between the four land cover types, the monthly data is simply accounting for observed variability across the period of measurement at each site. These methods will be clarified in the text.

Figure 5 As described, your p values are wrong. However, if you fix the analysis, please anyway only write p<0.05 (or whatever threshold you use), not p=.... This gives false confidence. Please consider changing to W/m2, which is used more often than MJ/m2/day. Please use only the common measurement period as the fluxes are likely not stationary. Fonts too small (use the caption font as an indicator of an appropriate size). Did you check what the low values of longwave radiation represent (10–15MJ/m2/day)? Are these real observations or issues with the instruments?

R: We will revise the p values in the text and figures to be expressed as either greater than or less than the alpha of 0.05. Font sizes in this figure, as with all other figures, will be increased in the revised manuscript to improve clarity. We thank the reviewer for advising we check the low values of longwave radiation. The issue has been identified and will be remedied. We chose to express our energy flux data in units of MJ/m2/day. To address our objectives, we prefer this method of expressing radiation data rather than W/m2. The MJ/m2/day totals better suit our radiation balance approach (accounting for inputs and outputs) in our conceptual framework and is a manner consistent with fundamental surface climate textbooks (e.g. Oke, 1987; Bailey, Oke, & Rouse, 1997).

545–563 Given the small distance between the sites, incoming radiation is not very interesting. Please analyse albedo instead. It would also be interesting to estimate

snowmelt timing at the different sites and analyse that. In particular as you discuss albedo later.

R: We thank the reviewer for this recommendation. Albedo varies only slightly between sites (0.05 to 0.19) whereas incoming solar radiation varies by a factor of 2 between the end-members of dense forest and open (treeless) wetland. We will add further explanation of this in the revised Introduction section (see response to 519 – 520).

Section 3.2.3 Do you have new numbers or results to add to this literature review? What do you mean by 'runoff', only in streams/rivers or also as groundwater?

R: This section will be revised and shortened. The purpose of this section is to provide interpretation of how the land cover stages differ from one another based on their hydrological function. For this we drew upon the large number of hydrological studies conducted on each of the land covers represented in Figure 4. The use of the term "runoff" was clarified and refers to the fraction of hydrological input from the atmosphere that does not remain in storage but follows hydrological pathways to the basin drainage network (channel fens) and from there on-ward to the basin outlet.

617–620 How about mosses?

R: We were not clear on what the reviewer was asking in this case, however, our revised section on hydrological function associated with each cover type presented in Figure 4 will include greater explanation of the impacts of vegetation changes (including mosses and other non-vascular plants) on evapotranspiration.

666–667 Does landcover depend on climate or climate on landcover?

R: This paragraph will be rewritten. To clarify this specific statement, this sentence will be revised as follows: "Coupling a broad-scale mapping initiative with the detail of site-specific data collected in the Scotty Creek basin demonstrates a permafrost thaw induced land cover transition."

667–669 Did you really show how the changes were initiated?

R: This sentence will be removed.

  675–678 Where exactly can I find these results?

R: This sentence will be removed. In the revised Conclusions section, the contributions of this paper and those that are synthesised within this paper for the purpose of interpreting the hydrological implications of land cover change, will be clarified.

681–684 Here again you mention that your research was on permafrost distribution. However, you did not analyse permafrost distribution. If you prefer to keep the statements on permafrost, you need to make it more clear how you measure or estimate permafrost distribution.

R: We will remove the word "distribution" from the manuscript as used in this sense.

Specific comments

* It would have been more convenient if you used hyperlinks so I could click on the references.

R: There was no requirement or guidance for this in the formatting instructions for manuscript submissions. Hyperlinks will appear in the final product as part of the production editing process.

183 'which are not only found extensively': the studies or the collapse scars? Maybe rephrase the sentence.

R: This sentence refers to the peat plateau and collapse scar wetland landscapes being found extensively throughout northwestern Canada and the global subarctic. The sentence will be revised as follows to clarify this: "The Scotty Creek drainage basin occupies one of many peatland-dominated lowlands of the Taiga Plains, and as such its landscape is dominated by complexes containing tree-covered peat plateaus overlying permafrost alongside treeless and permafrost free collapse scar wetlands. Such complexes are separated by channel fens that collectively function as the basin drainage

network (Hayashi et al. 2004; Quinton et al. 2009). This type of land cover not only dominates the lowlands of the Taiga Plains but is also found extensively throughout northwestern Canada and across the circumpolar subarctic (Olefeldt et al. 2016)

221 Start a new paragraph for this new thought.

R: A new paragraph will be created, as suggested, to separate this topic.

238 describes

R: This will be corrected in the revised manuscript.

237–244 These two sentences are very long and complicated. Please split them into more sentences.

R: These long sentences will be removed and replaced with more concise text.

251 With 'this study', do you mean your current paper, or Quinton 2019?

R: We will clarify this sentence to reflect that it refers to 'the present work'.

252–253 Sentence a bit confusing. How about 'We used precipitation data of the years ... to ... collected by the SCRS.'

R: For clarity, this sentence will be removed and replaced with: "In addition, we examined the precipitation data collected from 2008 to 2019 (Geonor, Model T200B) in relation to the three hydrological components listed above to gain insights into how changes in land cover stage affect the land cover water balance."

569 The word 'plotted' here and at lot of places in the manuscript irritates me. Try to omit it.

R: We will reduce the number of instances of the word 'plotted' throughout the manuscript. However, there are some sentences where 'plotted' is the most appropriate term and cannot be sufficiently substituted without altering the meaning of the sentence.

---

## Author Response (AR2)

**Key Changes:**

1. The manuscript was given a new title to specify this work relates to climate-induced landcover changes rather than encompassing all successional landcover change unrelated to climate change.
2. The Abstract and Key Points were revised to better integrate our methods and results.
3. The Introduction was completely rewritten. This allowed us to more clearly define our objectives, better introduce our conceptual framework, and describe how our work fits within the existing body of knowledge.
4. The Methods section was completely restructured and largely rewritten. This allowed us to better connect the conceptual framework with the work completed in this study.
5. Additional references were added throughout to better frame our work within the broader field.
6. All existing figures were updated and two new figures were created for the revised manuscript.
7. Much of the Results and Discussion section was rewritten to clarify what information was a new result and what information was synthesized from existing literature.

**Please see below for detailed responses to all reviewer comments.**

**REVIEWER #1**

**Review on**

*The trajectory of landcover change in peatland complexes with discontinuous permafrost, northwestern Canada*
Olivia Carpino, Kristine Haynes, Ryan Connon, James Craig, Élise Devoie and William Quinton

**Short summary**

The authors describe a conceptual model of landscape development from a permafrost underlain forest to a treeless wetland and, as last step, an afforested wetland. This conceptual model is underlain with historical and recent aerial photographs and energy and water balance data of one field site to describe the transformation in more detail. The study is motivated by a large scale analysis of the spatial distribution of landcover types in northwestern Canada.

**General comments**

This study needs more effort on the concept, the analysis, and the writing. My major critique on the concept is that the large scale analysis is not linked well with the conceptual model. What does the conceptual model mean at a larger scale? What is the timeframe when we would expect such changes? How big of an area is likely to change when? I am missing the space-for-time substitution that is mentioned in the abstract. This would improve the scientific significance. At the current stage, it is not clear what the new contribution of the study is.

Parts of the analysis itself are questionable, sometimes because they are just not well enough described. The statistical analysis with ANOVA cannot be used for auto-correlated data (such as the monthly values in this case); also comparisons should always be limited to the common period as with climate change most variables

are certainly not stationary. Changes in permafrost area are mentioned in several places, but it is not clear how the permafrost area was estimated.

The writing needs to be more specific on what the authors did for the current study. In multiple places of the paper it is hard to distinguish between their work and other peoples work. The paper would be much easier to read if they used the active voice for everything they did and found out. It is also important that they separate the results from the discussion. That would help a lot to distinguish what the new contribution in this study is as compared to previous understanding and the literature cited. This is something that needs to be highlighted. In the current version, the joined section reads like a literature review in lots of paragraphs. Even the methods section includes parts that should be moved to the discussion or introduction. The description of the methods is, in many places, not clear and for some of the described methods it is not clear to me which results they generated. The complete methods section should be restructured (suggestion below) and the remote sensing methods should be illustrated with a figure. In several parts I am also missing information on why a specific method/dataset was used. The English is fine but the quality of the figures could be improved. The complete paper is much longer than it would need to be to address the objectives; it would be better is it was more concise.

**Response (R): We thank the reviewer for these suggestions. We appreciate the detailed and thorough review. The introduction section has been completely rewritten and the methods section has been largely rewritten and restructured to address some of these concerns. These changes provide more context for the study and improve the linkages between the conceptual framework (which is no longer referred to as a conceptual model in response to this comment and others) and analysis. We also clarify the new contributions of this study.**

**Specific comments**

Abstract I waited until the last sentence to learn about the main method of this paper and I find it difficult to extract the main result and the main message from the abstract. I suggest to put the information about the method right after the first topic introduction sentences, be more specific in the results sentences and add a statement about the implication of this work.

**R: As suggested, the abstract has been revised and restructured in order to first introduce the topic of the paper, identify the methods used and present some of the key findings of the work. We have clarified the major results of the research and, through the restructuring of the abstract, clarify its overall message.**

18 'This study explores the current trajectory of landcover change across...' this is what I would like to see in the study. However, the large scale analysis is quite disconnected from the rest.

**R: We have clarified our approach in the abstract by expanding on the sentence identified by the reviewer here. We also provide more detail as to how the broader geomatics data spanning the Taiga Plains is integrated with our field-based hydrometeorological data collected in the Scotty Creek basin, located within the Taiga Plains. The large-scale change in forested peatland cover observed across the latitudinal, and therefore climatic, gradient of the Taiga Plains is reflective of observations of localized change from permafrost plateau to collapse scar wetland and ultimately afforested wetland occurring in the Scotty Creek basin.**

19 Where are you doing a space for time substitution? Mostly, you use a single location (Scotty Creek) as a substitution for a large area and show how it evolved over time.

**R: The 600 km north-south latitudinal gradient of the Taiga Plains also represents a gradient in prevailing climate. Therefore, the broad-scale change in forested peatland cover across this climatic gradient is representative of the anticipated landscape changes in a localized area (represented by the Scotty Creek basin) within that gradient over time. The forested plateaus in the northern Taiga Plains and the afforested wetlands at the southern region of the Taiga Plains frame the endmembers of landscape transition. These endmembers as well as plateau-wetland complexes are all located within the Scotty Creek basin. Therefore, our knowledge of the hydrological and thermal mechanisms governing these landcover types from field-based research in the Scotty Creek basin provide us with an understanding of the mechanisms contributing to landscape change. These results can then be integrated with the geomatics results across the Taiga Plains to anticipate the trajectory of change in the discontinuous permafrost zone. We further clarify this connection and the space-for-time substitution both in the abstract and in the Methods section of the revised manuscript.**

22–24 This needs to be included in Figure 3.

**R: To clarify the concurrent changes in wetland coverage, labels have been added to the figure detailing the dominant wetland type occurring with the changing forested area with latitude. At the high latitudes, "Forest with permafrost" has been added, while "Collapse Scars" and "Permafrost-free Forest" have been added at the mid- and low-latitudes, respectively.**

Key points The key points are all about methods. I suggest to include at least one on your findings.

**R: We have edited the Key Points to mention our methodological approach in one point as well as include two points on our findings. Key Points read as follows in the revised manuscript:**

1. **Conceptual framework developed to understand the trajectory of permafrost thaw-induced land cover change**

2. **Permafrost thaw-induced land cover changes vary latitudinally across the plateau-wetland complexes of the discontinuous permafrost zone**

3. **Partial wetland drainage triggers ecohydrological and thermal feedbacks that promote reforestation after full permafrost thaw**

Introduction The introduction touches on many interesting points. However, I have trouble to follow the introduction as the paragraphs do not seem to have one clear focus each and build on each other. Maybe you could slightly reorder the sentences and start every paragraph with a topic sentence, for example introducing current landscapes, observed changes, implications for the water and energy budget. The current last paragraph is very helpful.

**R: We thank the reviewer for this comment. The Introduction has been entirely re-written to improve the focus.**

40 Please update this reference to a peer-reviewed paper which also includes the thaw component.

**R: The following peer-reviewed reference has been added to replace the original reference:**

**"Box, J.E., Colgan, W.T., Christensen, T.R., Schmidt, N.M, Lund, M. … Olsen, M.S. (2019). Key indicators of Arctic climate change: 1971-2017. Environmental Research Letters, 14, 045010. https://doi.org/10.1088/1748-9326/aafc1b"**

43 What is 'not well understood'? Two sentences later you write that lots of changes have already been documented. Please be more specific on the lack of knowledge.

**R: The introduction has been entirely rewritten to improve clarity.**

92–98 This part of the paragraph seems to belong to the second paragraph of the introduction.

**R: As requested, these lines of text have been moved to the end of the second paragraph of the Introduction.**

Figure 1 All fonts are too small, some are barely readable. Please use vector graphics (such as pdf) for all figures so that the resolution is high and the text does not appear blurry. What does the yellow line mean? Can you indicate with keywords what it separates? How was the border between the permafrost regions determined? Are you using the map by Brown et al.?

**R: Font sizes have been increased and the figure has been exported at a higher dpi resolution to improve the clarity of the image. We added the label of "Taiga Plains ecoregion" with an arrow pointing to the yellow line. We added a sentence to the figure caption that the boundary between the sporadic and continuous is from Brown et al. (2002).**

Section 2.1.1 One or two pictures (maybe as part of Figure 1) would be good to show the different landscape parts. Please indicate where permafrost can be found.

**R: An additional figure (Figure 3 in revised manuscript) has been added to show the different land covers described.**

167–169 Can you specify how much the temperature increased at this site specifically?

**R: To clarify this point, the following text has been added:**

> **"Data collected by Environment and Climate Change Canada at the Fort Simpson A climate station show that MAAT has increased by approximately 0.05ºC/year since 1950, with warming most pronounced during the winter."**

179–181 'relatively long record', 'Long-term observations': how many years? Are those continuous measurement series?

**R: This section has been rewritten. The length and nature of the Scotty Creek observations has been clarified with the following revised sentences:**

> **"Scotty Creek (61.3ºN, 121.3ºW) has been the focus of field studies and monitoring since the mid-1990s and as such, the long-term and detailed data archive at Scotty Creek (Haynes *et al*., 2019) provide a unique opportunity to evaluate land cover changes over a period that coincides with rapid climate warming. Scotty Creek therefore also provides a reference to interpret land cover changes for terrains that are also present throughout the region."**

**Details on the length of observations used in this study can be found in the revised manuscript on lines 325-326, 342, 347, 363-364, 379-380, 384-387.**

Section 2.2 is not very clear to me. The methods could be described more clearly and I would like to read some sentences on why a certain wavelength/dataset/... was selected. The section would also profit a lot from a figure showing a small example area in all the different datasets and computed products. A table would also help, stating the most important properties for each dataset such as spatial coverage, resolution, date of acquisition, categories contained, who created it, citation. This could also be moved to the appendix.

**R: Much of this section has been rewritten and reorganized in order to better clarify the methods described in this manuscript. A figure (Figure 2 in revised manuscript) was also be made to illustrate a sample area of each dataset. Further information on each dataset has been added in-text including information on spatial coverage, resolution, data categories, etc.**

194 How is the warm season defined? How can you exclude moisture variations? I assume that you have different acquisition dates and some may be after a rainfall.

**R: Yes, the 70 Landsat scenes used in this mosaic have varying acquisition dates. Soil moisture variations were not impactful on classifying forest cover, which was the purpose of the Landsat mosaic. The Landsat methods have been clarified in the manuscript. The sentence referencing soil moisture variations has been removed to avoid confusion. The warm season is defined as the snow-free period. This was also be clarified in the manuscript by replacing "warm season" with "snow-free season".**

189–198 Did I understand it correctly, that you selected one image per scene based on fewest possible clouds, latest possible year, and month in June/July/August? Maybe you could make the collection criteria more clear. Concerning vegetation development, the beginning of June is quite different as compared to mid of August. Can you justify the 'rendering the images seasonally comparable' a bit better?

**R: Yes, that is correct. As the main purpose of the Landsat imagery was to identify black spruce-dominated forests, the absence of snow was the main priority. Other vegetation development is much more variable between June and August. The sentences beginning at line 194 in the original manuscript were rewritten to read:**

> **"Acquiring imagery during the snow-free season was prioritized and as such, all 70 Landsat tiles were acquired in June, July, or August, rendering the coniferous forest cover seasonally comparable and allowing for a more streamlined mosaicking process."**

198–199 Why did you restrict yourself to those 3 bands? Can you explain why you did not include more?

**R: The Landsat data was used for the purpose of classifying forest cover within identified peat plateau wetland complexes. As such the three bands used were deemed sufficient. Additional bands may have been useful had the Landsat data been used in another way. Near infrared (red), red (green), green (blue) (Landsat 8 bands 5, 4, 3) is a traditional band combination useful for visualizing vegetation characteristics.**

205–207 I do not know this dataset. Can you describe it briefly (What variables? Continuous or in classes? Spatial resolution? Vector or raster data?) Did you apply thresholds? Please cite a documentation and not only the download link.

**R: The following sentences have been added to the manuscript to provide further details:**

> **"The saturated soils dataset is part of a larger digital cartographical project by Natural Resources Canada, CanVec. The CanVec dataset is a vector format dataset, which can be downloaded by province/territory or Canada-wide and includes over 60 features organized into 8 themes, including land features. Land features, including the distribution of saturated soils, were digitized at a scale of 1:50000 (Natural Resources Canada 2017)."**

207–209 I also do not know this dataset and the reference does not appear in the literature list. Can you describe the dataset briefly (What variables? Continuous or in classes? Spatial resolution? Vector or raster data?) Did you apply thresholds? Please cite a documentation.

**R: We thank the reviewer for bringing this missing reference to our attention. The reference to this dataset has been added. Two additional documentation references were also added. The following sentences have been added to the manuscript to provide further information on this dataset:**

> **"The NCSCD is also a polygon database developed by the Bolin Centre for Climate Research through synthesizing data from numerous regional and national soil maps alongside field-data collected across Canada, USA, Russia, and the EU. The NCSCD includes data on the fractional coverage of different soil types and stored soil organic carbon (Hugelius et al. 2013a; Hugelius et al. 2013b). While the original format of the NCSCD is a vector, gridded data is also available at resolutions varying from 0.012° to 1° for circum-arctic use (Hugelius et al. 2013b). In addition to the circum-arctic dataset, the NCSCD is also available on a country-wide or regional scale, including a Canada product (Hugelius et al. 2013b)."**

**The added references are as follows:**

**Bolin Centre for Climate Research (2013). The Northern Circumpolar Soil Carbon Database. Available at: https://bolin.su.se/data/ncscd/ (Accessed March 20, 2019)**

**Hugelius G., Bockheim J.G., Camill P., Elberling B., Grosse G., Harden J.W., Johnson K., Jorgenson T., Koven C.D., Kuhry P., Michaelson G., Mishra U., Palmtag J., Ping C.-L., O'Donnell J., Schirrmeister L., Schuur E.A.G., Sheng Y., Smith L.C., Strauss J. and Yu Z. (2013a). *Earth System Science Data*, 5, 393–402. DOI:10.5194/essd-5-393-2013**

**Hugelius, G., Tarnocai, C., Broll, G., Canadell, J. G., Kuhry, P., and Swanson, D. K. (2013b)** *Earth System Science Data*, **5**, 3–13. DOI:10.5194/essd-5-3-2013

203–211 This step seems central to me and it is very abstract. It would be great to see a figure with examples of the datasets in combination with the Landsat imagery and the result of your filtering. One small area in all the different images.

**R: The new figure illustrating the methods (Figure 2 in revised manuscript) mentioned above includes this.**

212–214 It is not clear to me what you clustered here. Is it 3-band-Landsat pixels?

**R: The unsupervised Iso Cluster classification was done on the Landsat mosaic.**

215–217 How did you aggregate the classes? Manually based on expert knowledge? What was the advantage of having the unsupervised clustering first, if you targeted the specific classes described?

**R: Yes, the aggregation was completed manually. The unsupervised approach is an effective approach due to the large latitudinal span as well as the classification being completed on areas already identified as likely peat plateau-wetland complexes. Unsupervised classifications have the benefit of maximizing the number of classes, which can be especially useful if unexpected or uncommon classes are found. Manual aggregation is very commonly performed on unsupervised classifications. When using manual aggregation, first-hand knowledge of the area or imagery can then be used to aggregate the spectral classes into classes that are more meaningful to the specific study. Further text has been added throughout the geomatics methods to clarify the approaches used.**

220 Which map of peatland distribution?

**R: The map of peatland distribution referenced is the peatland distribution identified in the previous paragraph, which was developed using the saturated soils dataset from NRCan and the NCSCD. This has been clarified by introducing this earlier in the same paragraph.**

212–221 Can you add the clustering result to the figure I suggested?

**R: A new figure (Figure 2 in revised manuscript as mentioned above) has been added to the geomatics section.**

228–229 What do you mean by 'This generated a spatially distributed dataset'? I thought you reduced the spatial dimension to a north-south gradient.

**R: That is correct. The words "spatially distributed" have been removed.**

Section 2.3 is long and confusing. It should be restructured and some parts should be move to the introduction or discussion. It should contain only references to specific methods/datasets and not general findings on landscape change. I suggest to arrange the complete methods section as follows:

1. The Taiga Plains ecozone

2. The Scotty Creek field site

3. Geomatics methods

4. Water balance data (currently lines 234–237, 252–255, 260–282)

5. Imagery of the Scotty Creek basin (currently lines 283–295, 328–333)

6. Energy balance data (currently lines 299–323)

7. Conceptual model (currently lines 324–328, 333–340, needs more details)

**R: To improve the clarity of the descriptions of the study sites and methods, these sections has been re-organized as suggested. References to findings of landscape change that are not critical to the explanation and justification of the methods used in this study has been removed from the Methods section. Overall, the Study Sites and Methods sections has been restructured with the following headings in order to clarify our methodological approach:**

**2.      Study Site**

**2.1  The Taiga Plains Ecozone**

**2.2  Scotty Creek, Northwest Territories**

**3.      Methods**

**3.1  Geomatics Methods**

**3.2  Scotty Creek Imagery**

**3.3  Hydrological Data**

**3.4  Radiation Fluxes**

**The hydrological, radiation and land cover data used in the development of the conceptual framework (presented in the Results/Discussion sections) are each described in the appropriate Methods sub-section above. The conceptual model has been left to the Results/Discussion section.**

237–252 I do not see how these paragraphs fit into the section 'field based methods'. Please see my general comments.

**R: With these sentences, we are aiming to explain the relationship between the broad-scale landscape change occurring in the plateau-wetland complexes of the Taiga Plains and the localized permafrost thaw-induced changes observed within the Scotty Creek basin. With the restructuring of the Methods section, this explanation is no longer associated with the "field-based methods". Instead, the "field-based methods" section has been further divided into section 3.2, 3.3, 3.4 as noted above. Additionally, we have edited these statements to clarify the connection between the large-scale trends and our observations of localized landcover and hydrometeorological change in the Scotty Creek basin.**

253 'Interannual': please specify which years.

**R: The word 'interannual' has been removed and the precipitation monitoring years of 2008 to 2019 has been specified.**

255–259 This belongs to the discussion not to the methods.

**R: These sentences have been removed from the Methods and the concept of increasing hydrological connectivity (as opposed to precipitation inputs) contributing to observed elevated runoff from the Scotty Creek basin has been mentioned in the Discussion where appropriate.**

260–262 Please indicate how runoff was measured.

**R: Discharge from Scotty Creek has been measured by the Water Survey of Canada since 1995. The gauged portion of the basin spans 152 km$^2$. This information has been provided in the revised manuscript.**

Table 1 The caption sentence 'Both..thaw' should be moved elsewhere (The methods section on Scotty Creek would be a good place). You mention a runoff increase above. For non-stationary processes, it is misleading to calculate residuals from variables averaged over different time periods. Please use the common period 2013-2016 to calculate residuals. The numbers from the complete datasets can be added as additional columns or rows.

**R: This sentence has been removed from the Table 1 caption and instead this information has been rewritten and incorporated into the overarching explanation of the development of the conceptual framework in the rewritten Methods sections.**

**The dynamic landscape of the Scotty Creek basin is changing over time as permafrost thaws as a result of climate warming. Therefore, the runoff time series essentially represents different transitional stages with different proportions of each land cover type comprising the overall basin landscape. Consequently, we are selecting representative runoff values associated with the different stages of landscape change (permafrost-dominated forest as compared to wetland-dominated and subsequently permafrost-free forest). We highlight this in the revised manuscript so as to clarify the reasoning for not utilizing the common time period of 2013-2015.**

277–278 Table 1 does not show annual values. I think annual values would be interesting to see the variability and showing them would answer to my comment on Table 1. Why do you not show a figure with the time series of annual precipitation, runoff, evapotranspiration, and residual storage which you used for your study.

**R: Table 1 shows annual values. Figure 4a (in the original results/discussion, now Figure 6a) shows how these components change over time, though precipitation is not included as no changes in annual precipitation have been observed. The Figure 4a caption (now Figure 6a in the revised manuscript) has been modified to indicate the connection to Table 1.**

280–282 Why do you analyse the runoff, evapotranspiration, and storage data for trends but not the precipitation data?

**R: Previous research in the Scotty Creek basin has observed virtually no change in total annual precipitation throughout the period of data collection (Connon et al. 2014; Haynes et al. 2018). Therefore, we chose not to display the precipitation trends in the water balance portion of the conceptual framework, opting rather to display only runoff, storage and evapotranspiration, which are hydrological indicators of landscape change in this environment. However, annual precipitation values are used in the calculation of the storage term as a residual. We have clarified the reasoning for not incorporating precipitation visually into the water balance of the conceptual framework in sub-section "3.3 Hydrological Data" of the revised manuscript.**

293–295 By you or by the other authors you cite above?

**R: The quantification of landcover changes in this study incorporates the aerial photo and remote sensing imagery classifications originally presented by Quinton et al. (2010), Carpino et al. (2018) and Disher (2020). However, our work takes these previous studies to present a comprehensive examination of landscape change in the Scotty Creek basin over the complete period of the imagery record (1947 to 2018). This has been clarified in the sub-section "3.2 Scotty Creek Imagery" of the revised manuscript.**

296 – 299 This does not belong in a methods section.

**R: This statement has been removed from the Methods section in the revised manuscript.**

318 Here you mention 'subcanopy' are all measurements below the canopy? Please specify this when you describe the stations.

**R: The following sentence has been added to the first paragraph of section 3.4 to clarify radiation measurements:**

> **"All radiation measurements were made below the tree canopy at a height of 2 m above the ground surface."**

320–323 What do you test with the ANOVA? What are your responses and drivers? Are you looking for the effect of station land cover on November reflected shortwave radiation, for example?

**R: The ANOVA tested whether or not there was a significant effect of landcover (driving variable) on monthly shortwave and longwave incoming and outgoing radiation (response variables). Changes in each of the energy components over time are not statistically tested in this study.**

328–333 What are these images used for as compared to the airborne and satellite images described in l. 283–295?

**R: The RPAS imagery is used to visually illustrate the stages of landscape change identified in our conceptual framework. These aerial images all collected in the Scotty Creek basin illustrate, with the use of photographs, the mosaic-like landscape in this region. In contrast, the aerial photos and satellite**

imagery are used to quantify the proportions of the landscape represented by each of the peat plateau, collapse scar and afforested wetland landcover types in the Scotty Creek basin over the period of available imagery (1947 to 2018). We have now clarified the distinction in the application of the RPAS imagery and aerial photos and satellite imagery in the sub-section "3.2 Scotty Creek Aerial Imagery" of the revised manuscript.

324–328, 333–340 Here you touch on the conceptual model you developed, but it is not clear to me how you did it. The methods section should describe how you did your analysis and why you used a specific method/dataset, but not why you study something in general (this part should be moved to the introduction or maybe partly to the discussion section). You do not need to mention here which figures you show later.

**R: In the new sub-sections pertaining to the water balance, energy balance and aerial imagery field data collection, we have removed references to the overarching motivation and concepts that provide the foundation of the conceptual framework. With the restructuring of the Methods section as explained above, we have clarified the development of the conceptual framework in the rewritten methods section. Additionally, in the associated sub-sections, we clarified how the hydrology, energy and RPAS imagery components are incorporated into the water balance, energy balance and quantification of landcover change in the conceptual framework.**

337–340 Whether or not your results can be extrapolated is a topic for the discussion section, not for the methods.

**R: The purpose of the text in lines 337-340 of the original manuscript was to explain the approach that we have taken of using ground-based observation at an intensively studied site (i.e. Scotty Creek) to interpret remotely-sensed ground surface changes over a latitudinal transect. To improve the clarity of this point, the methods used at Scotty Creek were broken down into three methods sections: 3.2, 3.3, 3.4. As suggested by the reviewer, we also now wait until the Results and Discussion section to suggest using Scotty Creek as a way to represent the broader region. Some examples of this in the revised manuscript include: lines 412-414, 472-474, 479-483, etc. We also adjusted much of the text in sections 4.2.1, 4.2.2, 4.2.3 in response to this comment.**

Figure 2 What did you use the Landsat 8 data for in this map? As far as I understood, it was only used to estimate forest cover and not whether or not the landscape was 'peatland-dominated'. Please make this more clear in your methods! Some fonts are too small.

**R: We thank the reviewer for noticing that Landsat 8 data was included in the caption, this has been removed. Landsat 8 data was only used to estimate forest cover rather than predict the distribution of peatland-dominated terrain. Fonts have been increased on figures throughout the manuscript.**

344–345 Is this a finding, or a part of the original definition of a peatland which you used in the classification?

**R: This was a finding of the present study. The methods used are summarised in the caption for Figure 2. To clarify this point, the opening sentence of section 3.1 has been changed to the following:**

**"To place the analysis for Scotty Creek described above into a regional context, geomatics methods were applied to both zones of discontinuous permafrost within the Taiga Plains to quantify the areas occupied by each of the major land covers of all areas identified as peatland-dominated lowland."**

Figure 3 Please exchange the word 'proportional' with 'fractional'. You are not really showing 'proportional permafrost area', but only the rough classes. Are these from the Brown permafrost map? Do you have more detailed information? If not, please change the caption. The fonts are too small and the whole figure too big for the content. I would be interested to see an additional line for fraction of peatlands.

**R: "Proportional" has been replaced with "Fractional" and the fonts have been increased. As mentioned above, the dominant wetland type coincident with the changing forest cover at the high, mid- and low latitudes has been annotated in this figure (now Figure 5). The referenced Brown permafrost map was used, and this information has been added to the figure caption.**

365 'wetland features, including collapse scar bogs, are most prevalent' - This would be interesting to show in Figure 3

**R: To clarify the concurrent changes in wetland coverage, labels have been added to the figure detailing the dominant wetland type occurring with the changing forested area with latitude. At the high latitudes, "Forest with permafrost" has been added, while "Collapse Scars" and "Permafrost-free Forest" have been added at the mid- and low-latitudes, respectively.**

Section 3.2 It is not clear to me what the new part in this study is as compared to previous understanding and the literature cited.

**R: The Introduction has been re-written, and new preamble has been added before the conceptual framework is presented. This new text clarifies what is new and what is synthesised from existing work.**

396–401 Is this something you found out, or is it described in literature? Please cite one or more relevant articles and explain why you adopted/changed the phases.

**R: These stages of land cover change form the basis of our proposed conceptual framework for permafrost thaw-driven land cover change in peatland dominated regions of thawing permafrost. This has been explained much more clearly as a result of the revised Introduction and new text preceding the presentation of this new framework.**

418 You mention, that the transition is very fast (40 years). Please discuss speed in a bit more detail. Are there other studies? It would be good to add a rough timeframe in your Figure 4. The work of Claire Treat may be relevant here, e.g. Treat CC, Jones MC. Near-surface permafrost aggradation in Northern Hemisphere peatlands shows regional and global trends during the past 6000 years. The Holocene. 2018;28(6):998-1010. doi:10.1177/0959683617752858 (maybe other papers of her are even more interesting). She includes afforested peatlands in her work, but the timescales for forest recovery were more like 450 – 1500 years.

**R: Unlike traditional concepts of land cover change in peatland dominated regions of discontinuous permafrost in which forest re-establishment occurs over several centuries and is constrained by the rate of permafrost re-development, the concept presented here describes forest re-establishment as resulting from continued permafrost thaw, a process which removes "permafrost dams" and allows wetlands to de-water sufficiently for tree growth. This process occurs much more rapidly (within half a century) since the development of surface conditions that are sufficiently dry to support trees occurs much more rapidly as a result of wetland de-watering rather than as a result of permafrost re-establishment. New text explaining this difference has been added to the Introduction, and to the Results and Discussion section. Claire Treat's work has also been referenced in the context of this study. We thank the reviewer for this suggestion.**

419–423 What makes you think it is unlikely? I would like to see more discussion here.

**R: This sentence has been removed.**

Figure 4 Fonts are much too small. Why does incoming shortwave radiation change? Is it measured below canopy? In this case please rename this variable. I do not understand why storage changes across the gradient. As I understand it, storage is not a flux (like runoff and evapotranspiration) and the storage change (which would be a flux as your other two variables) should be close to zero on a multiannual timescale. You do not mention the timeframe here.

**R: The font size has been increased. Incoming radiation is measured below the canopy. We added new text to the Methods section and to the Results and Discussion sections so that this is now clear. Insolation is therefore greatest over wetlands because they are treeless, and least on plateaus with a dense canopy cover. It has been explained that as permafrost thaws, the density of the overlying canopy decreases and as a result, the insolation transitions toward the value measured over wetlands.**

**From year to year the change in storage of the landscape is indeed near zero, however as we are discussing landscape evolution, the formation of talik features and collapse scars allow for more water to be retained on the landscape. As thaw progresses, more of the landscape becomes connected to the drainage network and contributes to runoff, reducing the overall storage capacity of the landscape. In this sense, the maximum storage capacity of the landscape changes as permafrost thaws and this is what is meant by changing storage across the gradient. This has been clarified in the text.**

459–460 'at the expense of permafrost' - how do you know? You do not describe soil temperature or ice content anywhere. Please be more specific when you talk about permafrost.

**R: The revised introduction section clarifies this point by drawing on the large number of studies that demonstrate that permafrost thaw in the study region results in a transfer of forest (i.e. peat plateau) to treeless wetland. We added more text to explain this permafrost thaw induced land cover change from permafrost (forest) to permafrost-free (wetland). The revised "Scotty Creek, Northwest Territories" section contains information on permafrost characterization (soil temperature and composition, ice content, active layer thickness).**

Section 2.2.2 Please change your statistical analysis to incorporate autocorrelation and to use only the common period of all measurements. Please also do not provide exact p values but restrict yourself to p<0.05 (or whatever threshold you use).

**R: Changes in radiation components over time are not being considered with our statistical approach. Rather, the one-way ANOVAs were performed to assess potential differences in each of the four radiation components across the different land cover types examined in this study.**

**The data from each meteorological station is never treated against itself, but is instead compared with the data from each of the other three stations. Changes in each of the energy components over time are not statistically tested in this study. We are testing for differences in each of the radiation components individually between the four land cover types.**

**In the revised manuscript, p values have been expressed in relation to an alpha of 0.05.**

519–520 Is this measured above or below canopy? Is there shading on the sensors? I do not see why (given the small distance) incoming radiation should be different. I suggest to show albedo instead. It is more interesting and has more implications on the energy partitioning within the vegetation. This comment of course also applies at your statistical analysis and Figure 5.

**R: The revised Methods section explains that the measurements were made below the canopy. Because of different canopy densities (sparse to dense) and because of adjacent forested (plateau) and treeless (wetland) terrains, insolation can vary widely over short distances. On the plateaus, albedo varies by less than 5%. The difference between plateaus and wetlands in terms of albedo is greater, but this difference is still small in comparison with the contrast in insolation between these two sites. The greatest contrast occurs in late winter while the snow cover is still present in the forests but absent in the adjacent wetlands. These points have been incorporated and expanded upon in the revised manuscript.**

529–531 It is not statistically sound to use ANOVA on a time series (of monthly values in this case). The reason is, that the values are highly auto-correlated. Therefore, you get a 'fake confidence' and the p values are wrong. Either (I) remove your statistical analysis including all p values, (II) use an appropriate methods to include autocorrelation, or (III) use data with no (or at least little) autocorrelation, such as annual values. You could also analyse all mean June values in one analysis, as June 1999 should not be correlated to June 2000. This would give you one p value per month.

**R: Changes in radiation components over time are not being considered with our statistical approach. Rather, the one-way ANOVAs were performed to assess potential differences in each of the four radiation components across the different land cover types examined in this study. Changes in each of the energy components over time are not statistically tested in this study.**

**With respect to the issue of potential autocorrelation, the data from each meteorological station is never evaluated against itself, but is instead compared with the data from each of the other three stations. Therefore, as we are testing for differences in each of the radiation components individually between the four land cover types, the monthly data is simply accounting for observed variability across the period of measurement at each site. These methods have been clarified in the text.**

Figure 5 As described, your p values are wrong. However, if you fix the analysis, please anyway only write p<0.05 (or whatever threshold you use), not p=.... This gives false confidence. Please consider changing to W/m², which is used more often than MJ/m²/day. Please use only the common measurement period as the fluxes are likely not stationary. Fonts too small (use the caption font as an indicator of an appropriate size). Did you check what the low values of longwave radiation represent (10–15MJ/m²/day)? Are these real observations or issues with the instruments?

**As suggested, we have revised the p values in the text and figures to be expressed as either greater than or less than the alpha of 0.05.**

**Font sizes in this figure, as with all other figures, have been increased in the revised manuscript to improve clarity.**

**We thank the reviewer for advising we check the low values of longwave radiation. The issue has been identified and has been remedied. The low values noted by the reviewer were the product of an issue with the wetland station. The wetland station was not recording between Sept. 21, 2017 and Apr. 10, 2018. When the station was restarted, a new program was deployed that logged data at an hourly timestep instead of a half-hourly timestep. The R code used to generate the figure calculated cumulative radiation for one day, under the assumption of 30 min data intervals, which produced the low values (i.e. half of what we would expect). The R code has been modified to account for the programming shift at the wetland station so the values presented in the revised manuscript are now correct.**

**We chose to express our energy flux data in units of MJ/m²/day. To address our objectives, we prefer this method of expressing radiation data rather than W/m². The MJ/m²/day totals better suit our radiation balance approach (accounting for inputs and outputs) in our conceptual framework and is a manner consistent with fundamental surface climate textbooks (e.g. Oke, 1987; Bailey, Oke, & Rouse, 1997).**

545–563 Given the small distance between the sites, incoming radiation is not very interesting. Please analyse albedo instead. It would also be interesting to estimate snowmelt timing at the different sites and analyse that. In particular as you discuss albedo later.

**R: We thank the reviewer for this recommendation. Albedo varies only slightly between sites (0.05 to 0.19) whereas incoming solar radiation varies by a factor of 2 between the end-members of dense forest and open (treeless) wetland. We added further explanation of this in the revised Introduction section (see response to 519 – 520).**

Section 3.2.3 Do you have new numbers or results to add to this literature review? What do you mean by 'runoff', only in streams/rivers or also as groundwater?

**R: This section has been revised and shortened. The purpose of this section is to provide interpretation of how the land cover stages differ from one another based on their hydrological function. For this we drew upon the large number of hydrological studies conducted on each of the land covers represented in Figure 4 (now Figure 6). The use of the term "runoff" was clarified and refers to the fraction of hydrological input from the atmosphere that does not remain in storage but follows hydrological pathways to the basin drainage network (channel fens) and from there onward to the basin outlet.**

617–620 How about mosses?

**R: We were not clear on what the reviewer was asking in this case, however, our revised section on hydrological function associated with each cover type presented in Figure 4 (now Figure 6) made sure to include greater explanation of the impacts of vegetation changes (including mosses and other non-vascular plants) on evapotranspiration.**

666–667 Does landcover depend on climate or climate on landcover?

**R: This paragraph has been rewritten. To clarify this specific statement, this sentence has been revised as follows:**

> **"Coupling a broad-scale mapping initiative with the detail of site-specific data collected in the Scotty Creek basin demonstrates a permafrost thaw induced land cover transition."**

667–669 Did you really show how the changes were initiated?

**R: This sentence has been removed.**

675–678 Where exactly can I find these results?

**R: This sentence has been removed. In the revised Conclusions section, the contributions of this paper and those that are synthesised within this paper for the purpose of interpreting the hydrological implications of land cover change have been clarified.**

681–684 Here again you mention that your research was on permafrost distribution. However, you did not analyse permafrost distribution. If you prefer to keep the statements on permafrost, you need to make it more clear how you measure or estimate permafrost distribution.

**R: We removed the word "distribution" from the manuscript as used in this sense.**

**Specific comments**

* It would have been more convenient if you used hyperlinks so I could click on the references.

**R: There was no requirement or guidance for this in the formatting instructions for manuscript submissions. Hyperlinks should appear in the final product as part of the production editing process.**

183 'which are not only found extensively': the studies or the collapse scars? Maybe rephrase the sentence.

**R: This sentence refers to the peat plateau and collapse scar wetland landscapes being found extensively throughout northwestern Canada and the global subarctic. The sentence has been revised as follows to clarify this:**

**"The Scotty Creek drainage basin occupies one of many peatland-dominated lowlands of the Taiga Plains, and as such its landscape is dominated by complexes containing tree-covered peat plateaus overlying permafrost alongside treeless and permafrost free collapse scar wetlands. Such complexes are separated by channel fens that collectively function as the basin drainage network (Hayashi et al. 2004; Quinton et al. 2009). This type of land cover not only dominates the lowlands of the Taiga Plains but is also found extensively throughout northwestern Canada and across the circumpolar subarctic (Olefeldt et al. 2016)**

221 Start a new paragraph for this new thought.

**R: A new paragraph has been created, as suggested, to separate this topic.**

238 describes

**R: This has been corrected in the revised manuscript.**

237–244 These two sentences are very long and complicated. Please split them into more sentences.

**R: These long sentences have been removed and replaced with more concise text.**

251 With 'this study', do you mean your current paper, or Quinton 2019?

**R: We clarified this sentence to reflect that it refers to 'the present work'.**

252–253 Sentence a bit confusing. How about 'We used precipitation data of the years ... to ... collected by the SCRS.'

**R: For clarity, this sentence has been removed and replaced with:**

**"In addition, we examined the precipitation data collected from 2008 to 2019 (Geonor, Model T200B) in relation to the three hydrological components listed above to gain insights into how changes in land cover stage affect the land cover water balance."**

569 The word 'plotted' here and at lot of places in the manuscript irritates me. Try to omit it.

**R: We reduced the number of instances of the word 'plotted' throughout the manuscript. However, there are some sentences where 'plotted' is the most appropriate term and cannot be sufficiently substituted without altering the meaning of the sentence.**

**REVIEWER #2**

The manuscript entitled "The trajectory of landcover change in peatland complexes with discontinuous permafrost, northwestern Canada" by Olivia Carpino et al. focuses on the Taiga Plains of Northwest Canada, where rapid climate warming has significantly reduced the area underlain by permafrost in peatland complexes. A massive landscape shift has occurred in recent years, from a forest-dominated landscape to a wetland-dominated landscape. The authors explore the current trajectory of land cover change in a 300,000 km^2 area of discontinuous permafrost in northwestern Canada by presenting spatiotemporal variability using a 600-km latitudinal span of this region. By combining extensive geomatics data with ground-based meteorological and hydrological measurements, a new conceptual model of landscape evolution was developed.

This model explains the observed patterns of land cover change caused by permafrost thawing and provides a basis for predicting future changes. This is a very interesting paper and provides deep insights into how future permafrost loss may change the Taiga forest. The conceptual model and the discussion of water and energy balances are also interesting. However, I feel that the paper is somewhat disorganized and needs a more unified structure. I also think the conceptual model needs to be validated, and I would recommend that the validation be described in terms of the results and discussion. Overall, the authors need to revise the manuscript before its publication. Although there are some issues, I recommend that this paper be published after revisions are made.

**R: We thank the reviewer for these suggestions. We re-wrote the Introduction section, and added new text in the Results and Discussion section in advance of the presentation of the conceptual framework. These additions provide a better context for the framework. The sections that follow the introduction of the framework focus on the biophysical, hydrological, and micrometeorological functions of each land cover stage and the consequent changes to these functions as one stage transitions to the next.**

Specific comments

(1) Sections 3.1 and 3.2 of the Results and Discussion in Chapter 3 are in a completely different vein, making it difficult to read. Section 3.1 discusses the latitudinal distribution of forest cover and permafrost. I feel it would be better to show the percentage of peat areas and wetlands along with latitude in Figure 3. Similarly, the spatial distribution of forests and wetlands can be shown in a figure similar to Figure 2. Also, the spatial distribution is clearly shown in Figures 2 and 3, but the valuable aerial photos and data of IKONOS are described in the method of Chapter 2, which have been analyzed since 1947. I would like to see a figure similar to Figures 2 and 3, one that shows the changes over time based on the data analysis.

**R: We agree that these two sections were entirely in a different vein. To address this, we re-wrote the Introduction and introduce new text at the beginning of the Results and Discussion section to provide better context for the conceptual framework (as explained above). The percentage of peatland does not change with latitude in Figure 3 (now Figure 5) since it is based only on the peatland-dominated lowlands identified in Figure 2 (now Figure 4). However, we added some annotations to Figure 3 (now Figure 5) to clarify the variation in the type of peatland that predominates over the latitudinal transition. Specifically, we added: "Permafrost-free forest", "collapse scars", "forest with permafrost". In the figure caption we indicate that "the latitudes where permafrost-free forest, collapse scars, and forest with permafrost are most prevalent are indicated".**

(2) At the beginning of Section 3.2, a conceptual model of landscape change associated with the thawing of frozen ground is provided in Figure 4. The conceptual model is introduced so abruptly that it feels as though it has not been validated. Therefore, I would like to see the conceptual model validated on the basis of the analytical data in section 3.1. I would like you to show the results of the verification of the proposed model on the whole study region, using the data from Section 3.1, although it is valid for Scotty Creek. In addition, the purpose of this study was to characterize end members and intervening stages of the landcover transition. Please indicate the end members in Figure 4.

**R: We agree that the conceptual framework was introduced too soon in the text without sufficient context and explanation in advance. However, the new Introduction section and new text in the Results and Discussion section that precedes the introduction of the conceptual framework addresses this issue. The revised text conveys that the conceptual framework forms the basis of subsequent discussion on the form and function of each land cover stage. We clarified that in addition to presenting new data, this manuscript also draws upon the accumulated knowledge of hydrological studies on each of the land cover types presented in Figure 4 (now Figure 6). As such, the conceptual framework provides a synthesis of previous research in order to interpret the land cover stages, their form and function, and the processes driving their transition.**

(3) There is no single designation for Scotty Creek; please clarify if Scotty Creek Research Station (SCRS) is the same as Scotty Creek, Scotty Creek basin or Scotty Creek watershed. I recommend that Scotty Creek be unified with SCRS or others. In addition, I feel that there needs to be a map of the meteorological and hydrological observations that are being made at SCRS. In particular, a description or table of the four component radiation observations is needed. In Figure 5, it is difficult to understand the changes in the four radiative components without a description of whether they are observations above the vegetation canopy or on the forest floor. Additionally, please show which stage of the conceptual model each letter (a)-(d) corresponds to.

**R: A basin map with location of instrumentation has been included to provide reference for the reader. SCRS has been removed and replaced with reference to the Scotty Creek basin. The Methods section has been revised so that it is clear where the sensors are located in relation to the tree canopy. The sensor type and name and location of the manufacturer were added to the text: "…using a CNR4 sensor (Kipp and Zonen, Delft, Netherlands).**

(4) Lines 106-113: Here the characteristics of the energy balance of the forest canopy are described. However, the difference between wetlands and forests is also evident in the water balance. For example, the amount of precipitation reaching the ground due to rainfall and snowfall interception is smaller in forests, making them more prone to drying out than in wetlands. I think it is important to describe this point as well.

**R: In the revised manuscript, we have emphasized the distinctions in hydrology for each of the dominant landcover types in the discontinuous permafrost zone. With this explanation, we highlight, as suggested, the influence of the black spruce canopy of partitioning hydrological inputs in each of the landcover types similar to our description of the energy dynamics.**

(5) Line 135: Dry peat has been mentioned, but I think it is necessary to mention rainfall interception by mosses and other factors as a cause (e.g., Price et al., 1997, J. Hydrol, Suzuki et al., 2007, HYP). The reason for the

dryness of the peat layer is that mosses are thought to play a major role in blocking rainfall. It would be useful to describe the underlying vegetation of the forest, and such descriptions should be added.

**R: Where mosses predominate, the ground surface is already saturated and the topography very flat, so blocking of precipitation does not have a major impact on the partitioning of precipitation input into runoff and storage. We added to the text the following reference that provides the results of extensive vegetation surveys for each land cover type presented in Figure 4 (now Figure 6).**

**Garon-Labreque MÉ, Léveillé-Bourret É, Higgins K and Sonnentag O. 2015. Additions to the boreal flora of the Northwest Territories with a preliminary vascular flora of Scotty Creek. Can. Field-Nat. 129, 349–67. dx.doi.org/10.22621/cfn.v129i4.1757**